# Understanding In-Context Learning in Transformers and LLMs by Learning to Learn Discrete Functions

**Satwik Bhattamishra**$^{\wedge\,\dagger}$ **Arkil Patel**$^{\vee}$ **Phil Blunsom**$^{\wedge\oplus}$ **Varun Kanade**$^{\wedge\dagger}$

$^{\wedge}$ University of Oxford $\quad^{\vee}$ Mila and McGill University $\quad^{\oplus}$ Cohere

## Abstract

In order to understand the in-context learning phenomenon, recent works have adopted a stylized experimental framework and demonstrated that Transformers can match the performance of gradient-based learning algorithms for various classes of real-valued functions. However, the limitations of Transformers in implementing learning algorithms, and their ability to learn other forms of algorithms are not well understood. Additionally, the degree to which these capabilities are confined to attention-based models is unclear. Furthermore, it remains to be seen whether the insights derived from these stylized settings can be extrapolated to pretrained Large Language Models (LLMs). In this work, we take a step towards answering these questions by demonstrating the following: (a) On a test-bed with a variety of Boolean function classes, we find that Transformers can nearly match the optimal learning algorithm for 'simpler' tasks, while their performance deteriorates on more 'complex' tasks. Additionally, we find that certain attention-free models perform (almost) identically to Transformers on a range of tasks. (b) When provided a *teaching sequence*, i.e. a set of examples that uniquely identifies a function in a class, we show that Transformers learn more sample-efficiently. Interestingly, our results show that Transformers can learn to implement *two distinct* algorithms to solve a *single* task, and can adaptively select the more sample-efficient algorithm depending on the sequence of in-context examples. (c) Lastly, we show that extant LLMs, e.g. LLaMA-2, GPT-4, can compete with nearest-neighbor baselines on prediction tasks that are guaranteed to not be in their training set.

## 1 Introduction

Transformer-based large language models (LLMs) have shown a remarkable ability to learn tasks in-context using a handful of demonstrations and without updating their weights (Brown et al., 2020). Demystifying this 'in-context learning' ability theoretically and empirically is an intriguing direction. Since real-world language data and tasks are hard to define precisely, in-context learning has been studied in various simplified yet well-defined settings (Chan et al., 2022; Hahn & Goyal, 2023).

Recently, Garg et al. (2022) proposed a framework to understand the in-context learning phenomenon, wherein the model receives a prompt $P_k = (\mathbf{x}_1, \mathbf{y}_1, \ldots, \mathbf{x}_{k-1}, \mathbf{y}_{k-1}, \mathbf{x}_k)$ containing a sequence of pairs of inputs and labels followed by a query input. Each input $\mathbf{x}_i \in \mathbb{R}^d$ is sampled from a distribution $D_X$ and is labeled by a function $f : \mathbb{R}^d \to \mathbb{R}$ sampled from a distribution of functions $D_{\mathcal{F}}$. The goal of the model is to accurately predict $f(\mathbf{x}_k)$. Unlike works that study in-context learning on LLMs, in this framework, the model $M$ is *trained from scratch* on various such prompts sampled according to distributions $D_X$ and $D_{\mathcal{F}}$. Thus, this framework could be understood as a meta-learning approach, i.e., the models are trained to learn learning algorithms. Garg et al. (2022) show that such trained Transformers perform well on a range of prediction and regression tasks on inputs in $\mathbb{R}^d$. Given the flexible and interpretable nature of the framework, several recent works (Li et al., 2023; Ahuja & Lopez-Paz, 2023; Bai et al., 2023) have adopted it to further understand in-context learning.

**Research Questions.** Multiple works (Bai et al., 2023; Ahuja et al., 2023; von Oswald et al., 2022) have demonstrated that Transformers can learn various classes of real-valued functions in such in-

---

$^{\dagger}$Corresponding Authors: satwik.bmishra, varun.kanade@cs.ox.ac.uk

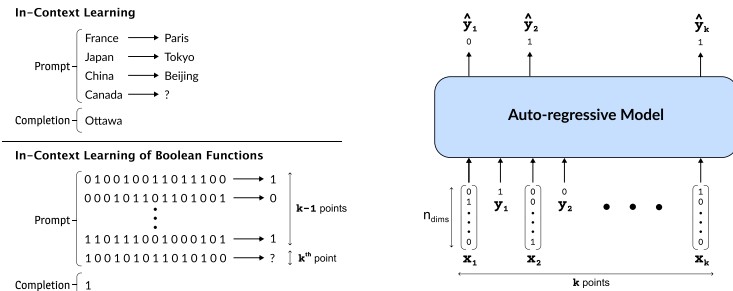

Figure 1: In-context learning of Boolean functions in our setup. In an autoregressive manner, given $k - 1$ points of the form $(\mathbf{x}_i, \mathbf{y}_i)$ where $\mathbf{x}_i \in \{0,1\}^n$ and $\mathbf{y}_i \in \{0,1\}$, a model has to learn to accurately predict the label $\mathbf{y}_k$ for the $k^{\text{th}}$ point.

context settings. This naturally motivates further questions: (a) What are the limits of the in-context learning ability of Transformers? (b) Is the attention mechanism essential for in-context learning? (c) Can Transformers exploit high-quality and informative examples to learn more efficiently? (d) To what extent do LLMs that are not specifically trained for these tasks carry the capacity to implement non-trivial learning algorithms on in-context examples?

In this work, we conduct an extensive empirical study with various classes of Boolean functions to make progress towards answering these questions. Since Boolean functions are widely studied in learning theory (O'Donnell, 2021), the sample complexity and learnability of various function classes are well understood which helps us in designing the experimental framework. Additionally, unlike real-valued functions, the discrete nature of the inputs allows us to adopt the setting for LLMs and test their ability to implement learning algorithms. We give an overview of our contributions below.

**In-context learning Boolean functions.** Transformers trained from scratch perform well on a range of Boolean function classes, however, their performance does degrade on seemingly more 'complex' classes. In particular, when learning parities their performance is as bad as random guessing, but known algorithms can perform near-perfectly with the same size dataset. We also evaluate the performance of recently proposed alternatives to Transformers which were motivated by the desire to have sub-quadratic inference. Our results indicate that while these models match the Transformer's performance on most tasks, there remain some gaps in others.

**Teaching Sequences.** For a class of functions, a teaching sequence is a sequence of examples that can uniquely identify the function. Our results show that Transformers have the capacity to learn more sample-efficient algorithms when trained on such highly informative sequences. For certain tasks, our results show that a single Transformer can learn two different algorithms and dynamically choose between them depending on the in-context sequence–it uses the sample-efficient version if the input has a teaching sequence, and falls back on the less sample-efficient algorithm otherwise.

**Investigation with LLMs.** The goal of recent work in stylized settings was primarily to understand whether LLMs can use in-context examples to learn novel tasks. However, it is still unclear if LLMs can implement learning algorithms rather than simply indexing from a list of tasks seen during pretraining. To understand whether LLMs can learn in a manner similar to models in the stylized setting, we investigate the performance of pretrained LLMs on the same tasks, by adapting them in two different ways. First, we use a frozen GPT-2 model, only training the input embedding layer, so as to be able to evaluate GPT-2 in the same stylized framework. Second, working with discrete inputs enables us to use the bit tokens directly to test the performance of LLMs by providing them in-context examples. Our results show that these models achieve non-trivial performance, competing with *nearest neighbor* classification. As we are sampling learning problems from a large combinatorial space, it is virtually guaranteed that these models are learning from in-context examples alone.

## 2 SETUP FOR IN-CONTEXT LEARNING

**In-context learning.** Our setup closely follows that of Garg et al. (2022) and is similar to other recent works which study the ability of Transformers to learn to implement learning algorithms in

their forward pass (Ahuja et al., 2023; Li et al., 2023; von Oswald et al., 2022). In this setup, a sequence model $M$ (such as Transformers) is trained using $N$ sequences, each sequence consisting of $m$ labeled examples, $(\mathbf{x}_1, y_1, \ldots, \mathbf{x}_m, y_m)$. The model is trained for the next token-prediction task, except that we only care about every alternate token: if $\hat{y}_1, \ldots, \hat{y}_m$ are the $m$ output labels predicted by the model, where $\hat{y}_k := M(P_k)$ is predicted using the prefix $P_k = (\mathbf{x}_1, y_1, \ldots, \mathbf{x}_{k-1}, y_{k-1}, \mathbf{x}_k)$, the loss on this sequence is given by $\frac{1}{m} \sum_{k=1}^{m} \ell(\hat{y}_k, y_k)$.

To generate the set of $N$ training sequences, we do the following: We sample $m$ points $\mathbf{x}_1, \ldots, \mathbf{x}_m$, i.i.d from some distribution $D_X$ over $X$, we sample a function $f \in \mathcal{F}$ from some distribution $D_{\mathcal{F}}$ over $\mathcal{F}$, and obtain the sequence $(\mathbf{x}_1, f(\boldsymbol{x}_1), \ldots, \mathbf{x}_m, f(\boldsymbol{x}_m))$. This is repeated $N$ times, with a fresh sample of $m$ points and a freshly sampled $f$. For an appropriate loss function $\ell$, the empirical loss over $N$ training examples is defined as,

$$\frac{1}{N} \sum_{i=1}^{N} \left( \frac{1}{m} \sum_{k=1}^{m} \ell(M(P_k^{(i)}), f_i(\mathbf{x}_k)) \right). \tag{1}$$

We will primarily work with classes of Boolean functions $f : \{0, 1\}^n \to \{0, 1\}$, which take as input a vector of bits of fixed length and output either $0$ or $1$. We say that a model $M$ can learn a class of functions $\mathcal{F}$ in-context if with high probability, for $f \sim D_{\mathcal{F}}$ and for a sufficiently large $k_{\mathcal{F}}$, for all $k \geq k_{\mathcal{F}}$,[1] $\Pr_{P_k \sim (D_X^k, D_{\mathcal{F}})}[M(P_k) \neq f(\mathbf{x}_k)] \leq \epsilon$.

We would like to highlight the difference from the traditional statistical learning framework, where a single target function $f \in \mathcal{F}$ is learned from a random sample of labeled examples. In our setup, each sequence is a *learning problem* and the model is learning a learning algorithm.

**Models.** We primarily focus on Transformers Vaswani et al. (2017) but we consider five different types of architectures for the majority of the experiments. These include a recurrent model (LSTM) (Hochreiter & Schmidhuber, 1997), a state-space model (DSS) (Gupta et al., 2022), a long convolutional model (Hyena) (Poli et al., 2023), and a hybrid model (RetNet) (Sun et al., 2023). We consider decoder-only GPT-like architectures containing causal masking (see Figure 1 right). Note that, each input $\mathbf{x}_k$ is a vector of dimension $n$. A linear map, $\{0, 1\}^n \to \mathbb{R}^d$, with learnable parameters, is applied to each input and label vector to map it to a vector of the same dimension as the model width. The models receive as input $(\mathbf{x}_1, \mathbf{y}_1, \ldots, \mathbf{x}_{m-1}, \mathbf{y}_{m-1}, \mathbf{x}_m)$ sequentially and for each prefix $P_k$ for $1 \leq k \leq m$, they predict the label $\mathbf{y}_k$ which is provided as the next token as in a language model.

**Training.** The models are trained from scratch to minimize the objective defined in Eq. 1 where cross-entropy is used as the loss function. The details of the hyperparameters and other aspects of implementation are provided in Appendix I. We have made our source code available at https://github.com/satwik77/incontext-bool.

## 3   IN-CONTEXT LEARNING BOOLEAN FUNCTIONS

In this section, we investigate the abilities of autoregressive architectures to learn various classes of Boolean functions in-context. Our experiments are geared towards answering the following questions: (1) What classes of Boolean functions can Transformers learn in-context and what are their limitations? (2) Is attention necessary for in-context learning and what are the differences between the capabilities of Transformers and attention-free architectures?

**Tasks.** We explore several classes of Boolean functions with varying characteristics. We define some representative classes here; the rest are defined in Appendix C.1.

**(i)** *Conjunctions and Disjunctions.* For the input domain $X_n = \{0, 1\}^n$, any input $\mathbf{x} \in X_n$ denotes a possible assignment to $n$ Boolean variables $x_1, \ldots, x_n$. A literal is either the variable $x_i$ or its negation $\bar{x}_i$. A conjunction is simply an *and* ($\wedge$) of some subset of the $2n$ literals. For example, for $n = 10$, one possible conjunction could be $x_2 \wedge \bar{x}_6 \wedge x_7$ which outputs 1 only when $x_2, x_7$ are 1 and $x_6$ is 0. The class of Conjunctions contains all such conjunctions over $\{0, 1\}^n$. The class of Disjunctions is defined similarly with the *or* ($\vee$) operation ($x_2 \vee \bar{x}_6 \vee x_7$).

---

[1]The choice of $k_{\mathcal{F}}$ may depend on the function class, as for more complex classes, more examples may need to be seen before learning is possible.

**(ii)** *DNFs and CNFs.* A Boolean function is in the class DNF if it is a disjunction of one or more conjunctions. Formally, for any $\mathbf{x} \in \{0, 1\}^n$, a function $f$ belongs to the class of DNFs, if it can be expressed in the form $f(\mathbf{x}) = (z_{1,1} \wedge z_{1,2} \wedge \ldots \wedge z_{1,k_1}) \vee \ldots \vee (z_{m,1} \wedge z_{m,2} \wedge \ldots \wedge z_{m,k_m})$ where $z_{i,j}$ are literals which are either the variables or their negation. Similarly, CNFs contain functions that are a conjunction of one or more disjunctions. While only the simplest of classification rules may be defined using conjunctions/disjunctions, a richer family becomes available when using DNF/CNF representations. Since DNFs/CNFs are conjectured to be computationally hard to learn, we consider 3-term DNFs and 3-clause CNFs which are known to be efficiently learnable.

**(iii)** *Parities* is one of the most fundamental classes of Boolean functions; widely studied in learning theory and cryptography because of its intriguing properties(Blum et al., 2003; Regev, 2010). A Parity function computes the XOR ($\oplus$) of some subset of the $n$ variables. Each Parity function outputs 1 when an odd number of the variables are assigned the value 1 and outputs 0 otherwise. The class PARITY-$n$ contains all $2^n$ possible Parity functions over $\{0, 1\}^n$. We denote the class of sparse parities with PARITY-$(n, k)$ which contain functions with $k$ relevant variables defined over $\{0, 1\}^n$.

**Sampling data and functions.** For each task such as Conjunctions or Parity, we sample a function and $m$ Boolean inputs to generate each training (or test) example. For the majority of the tasks, the inputs are sampled uniformly at random where each input bit is either 0 or 1 with probability $1/2$. However, for tasks such as Conjunctions and 3-term DNF the null accuracy under the uniform distribution will be close to 1, and hence we modify the input distribution to make the set more well-balanced. The details of the input distribution and the distribution over functions for each task are provided in Appendix C.2.

## 3.1 RESULTS

Table 1 summarizes the performance of various architectures on the Boolean function tasks considered in the paper. The numbers indicate the accuracy of prediction for the last example in a prompt sequence, averaged over 1280 sequences. The curves showing the accuracy as a function of the number of observed examples for all tasks are provided in Appendix C.4.

**Transformers show varied performance on Boolean functions.** We find that on certain *simple* classes of functions such as Conjunctions and Disjunctions, Transformers achieve perfect accuracy within a relatively small number of examples. We observe that the performance of Transformers is close to that of feedforward networks (FFN) trained with gradient-descent as well as known PAC-learning algorithms for the respective tasks (see Figure 13), in terms of sample complexity required to achieve perfect accuracy. On such tasks, we also find that they perform well on out-of-distribution inputs and functions; they also exhibit robustness to the presence of noisy labels during the training process. (see Appendix D.1 for more details.)

While Transformers perform well on certain classes of Boolean functions, we find that their performance degrades across others. The tasks can be broadly divided into three categories. The first category contains tasks where Transformers can achieve near-perfect ($> 95\%$) accuracy. The second category contains examples where the model performs better than baselines but its accuracy saturates below $95\%$. The last category consists of Parities and Sparse Parities where Transformers and other models do not improve beyond chance-level accuracy even with a large number of training examples. While prior works (Garg et al., 2022; Ahuja et al., 2023; Bai et al., 2023) have observed that Transformers perform well on a wide range of tasks, our results highlight that there are certain classes of functions where they are imperfect (second category) and there are certain classes such as Parities where they seem incapable of finding any learning algorithm to solve the task. Note that, for PARITY-$(10, 2)$, 140 examples are sufficient for feedforward networks trained with gradient-descent to achieve near-perfect test accuracy (see Figure 18). The tasks PARITY-20 and PARITY-$(20, 3)$ could be PAC-learned within 140 examples if the models could learn to apply the Gaussian elimination algorithm (Fischer & Simon, 1992). Hence, the failure of Transformers on the tasks in learning to learn Parities does not arise due to the lack of a sufficient number of in-context examples. We discuss and further explore the difficulty of learning Parities in Appendix F.

**Transformers vs. other Architectures.** For the majority of the tasks considered in this work, we observe that attention-free architectures perform almost as well as Transformers. The efficacy of recurrent and long-convolution models on multiple tasks such as Conjunctions and Disjunctions indicates that the in-context learning phenomenon in the sense of implementing learning algorithms

Table 1: The performance of different architectures on in-context learning Boolean functions. The numbers depict the accuracy of a model while predicting the last input in the prompt. For instance, for prompts with $m$ points, the number indicates the accuracy on the $m$-th input point based on the first $m - 1$ points and labels. The descriptions of the baselines are provided in Appendix C.3. For all values reported in the table, the standard deviation is within 0.5.

| TASK | N_DIMS | POINTS | NULL | AVERAGING | NN | FFN | DSS | LSTM | RETNET | HYENA | TRANSFORMER |
|---|---|---|---|---|---|---|---|---|---|---|---|
| Conjunctions | 28 | 70 | 69.9 | 68.2 | 81.3 | 94.0 | 97.2 | 100.0 | 99.4 | 100.0 | 100.0 |
| Disjunctions | 28 | 70 | 69.8 | 67.5 | 80.3 | 93.4 | 97.5 | 100.0 | 99.7 | 100.0 | 100.0 |
| Sparse Disjunctions | 50 | 70 | 63.9 | 70.4 | 71.7 | 91.2 | 99.5 | 100.0 | 98.2 | 100.0 | 99.9 |
| 3c-CNF | 28 | 100 | 59.3 | 66.6 | 79.6 | 91.5 | 93.3 | 94.3 | 91.5 | 95.5 | 95.1 |
| 3t-DNF | 28 | 100 | 55.6 | 70.3 | 80.4 | 90.9 | 94.7 | 95.5 | 92.9 | 96.0 | 95.2 |
| Majority | 28 | 100 | 55.5 | 80.9 | 67.8 | 84.0 | 90.7 | 91.2 | 89.6 | 91.5 | 91.5 |
| 0-1 Threshold | 28 | 70 | 50.5 | 70.7 | 77.1 | 89.6 | 72.4 | 83.9 | 72.9 | 85.0 | 85.3 |
| Integer-Halfspace | 20 | 70 | 51.4 | 81.7 | 70.0 | 90.6 | 84.0 | 84.6 | 83.1 | 86.9 | 86.4 |
| Parity-(10, 2) | 10 | 140 | 50.6 | 48.5 | 81.7 | 100.0 | 49.7 | 51.1 | 48.7 | 52.0 | 53.7 |
| Parity-(20, 3) | 20 | 140 | 50.0 | 50.7 | 55.3 | 50.2 | 51.2 | 50.0 | 51.1 | 47.6 | 49.1 |
| Parity-20 | 20 | 140 | 49.2 | 50.6 | 49.2 | 50.6 | 47.8 | 49.6 | 50.3 | 51.1 | 50.6 |
| Nearest Neighbor | 10 | 100 | 49.6 | 66.6 | 100.0 | N/A | 81.5 | 79.1 | 90.6 | 85.5 | 100.0 |

is not particular to Transformers. At the same time, we find that certain differences between Transformers and their recently proposed alternatives evidently exist. On tasks such as 0-1 Threshold functions, we find that RetNet and DSS perform worse than Transformers (see Table 1). With further experiments on linear regression tasks proposed in previous works, we find that recurrent models such as DSS and LSTMs struggle to match the performance of Transformers (see Figure 8 in Appendix). The closest to Transformers in terms of performance is Hyena, which matches Transformer's performance on all tasks apart from the nearest-neighbor (NN) task. The NN task tests the ability of models to implement the nearest-neighbor algorithm as described in Appendix C.1. Our results indicate that while recently proposed architectures with sub-quadratic inference time can perform in-context learning, there still exists a minor performance gap between these architectures and Transformers.

## 4 IN-CONTEXT LEARNING WITH TEACHING SEQUENCES

**Premise.** Our experiments in the previous section as well as previous works investigate the abilities of models to perform statistical learning from arbitrary distributions in an in-context setting. Given that we are exploring the ability of sequence models to learn 'learning algorithms', one could wonder: *Can sequence models find learning algorithms which are more sample-efficient when they are provided with a more informative sequence of examples?* We explore this question in this section.

**Teaching Sequences** (Goldman & Kearns, 1995). For a class of functions $\mathcal{F}$, the teaching sequence of a function $f$ in $\mathcal{F}$ is a sequence of labeled instances $(\mathbf{x}_1, \mathbf{y}_1, \dots, \mathbf{x}_m, \mathbf{y}_m)$ such that it unambiguously identifies $f$ in $\mathcal{F}$ – only the function $f$ is consistent with the sequence of labeled instances. We are interested to see if architectures such as Transformers can learn to identify the correct target function right after seeing the shortest teaching sequence[2] of that function.

**Experimental Setup.** Every aspect of the experimental setup except the data distribution remains identical to the previous experiments. For each sequence during training and evaluation, the first $t$ points are a teaching sequence of the target function and the next $m - t$ points are sampled from the distribution $D_X$. The sampling of functions, the training and the evaluation process remain the same. Although we primarily focus on Transformers, most results in this section hold for all architectures.

**Tasks.** We experiment with 5 tasks: Conjunctions, Disjunctions, CNFs and DNFs, and sparse parities. Every conjunction (or disjunction) has a teaching sequence of length $k + 2$ where $k$ is the number of

---

[2]From here onwards, by teaching sequence we will refer to the shortest teaching sequence of a function.

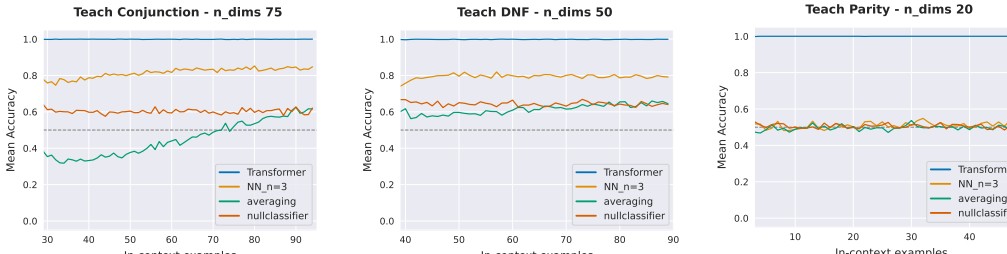

Figure 2: Performance of Transformers on various tasks with Teaching Sequences. The plots depict the performance of models right after the teaching sequence. Refer to Section 4 for more details.

literals in the function. Every 3-term DNF or CNF has a teaching sequence of length $l + 3$ where $l$ is the total number of literals in all 3 terms. The teaching sequence of each sparse parity function is of length $k$ where $k$ is the number of relevant bits upon which the Parity function is computed. For each task (such as Conjunctions), we refer to the version with teaching sequence as Teach <task-name> (e.g. Teach Conjunction). The details of the teaching sequences are provided in Appendix G.

**Transformers succeed in learning from Teaching Sequences.** For all 5 tasks, we find that Transformers achieve perfect accuracy on subsequent points after receiving the teaching sequence. Figure 2 shows the accuracy curve of Transformers and baselines on three representative tasks after receiving the first $t$ points containing the teaching sequence. Note that, the accuracy of Transformers stays (close to) $100\%$ after receiving the teaching sequence. It is interesting to see that Transformers succeed in learning with teaching sequences for tasks like DNFs and sparse parities where they struggled to learn in the vanilla setting.

By definition, the teaching sequence is the smallest sequence required to learn the target function. Hence, it can often be much smaller than the sample complexity for a learning algorithm to learn from arbitrary distributions such as the ones considered in the previous section. Thus, our experiments show that models predict perfectly with much fewer examples when provided with these teaching sequences. This is not true for FFNs trained with gradient descent which fail to predict accurately given only the teaching sequence during training (see Figure 18 right in Appendix). FFNs+GD is a more general-purpose algorithm and is hence not the most optimal for specific problems.

**Two distinct algorithms to learn one task.** An interesting finding is that Transformers seem to learn two different algorithms to learn Conjunctions depending on the data distribution $D_X$ during the training process. See that (Figure 3 left) when Transformers are trained to perform in-context learning with Conjunctions in the vanilla setting (as in Section 3) without teaching sequences, and tested on examples (or Prompts) containing teaching sequences, they are not able to leverage the teaching sequence and require more examples to predict accurately. The algorithm learned with standard Conjunctions still works on examples with teaching sequences even if it is not as sample-efficient. On the other hand, when Transformers are trained with teaching sequences and are tested in the standard setting without teaching sequences, they do not perform well since the learning algorithm relies on using the first $t$ examples containing the teaching sequence (Figure 3 center-left). These results indicate that models can learn two distinct algorithms depending on the distribution of inputs provided during training.

We conducted another experiment where we trained a Transformer on a mixture of examples with and without teaching sequence. During training, we sample an example (or Prompt sequence) with a teaching sequence with probability $\frac{1}{2}$, and without a teaching sequence with equal probability. We evaluate the model on both tasks separately, one which contains examples with teaching sequences and one without them. We find that the same Transformer model could achieve near-optimal performance for both tasks – when provided with teaching sequences, it behaves like the model that is trained on just the Teach Conjunction task (Figure 3 Right) and when provided with examples without teaching sequences, it behaves like a model trained on just the Conjunction task (Figure 3 center-right). This indicates that Transformers can learn two distinct learning algorithms for Conjunctions and implement the optimal one depending on the sequence of in-context examples. This highlights the versatility of neural sequence models in being able to find separate learning algorithms with respect to the data distribution for solving the same task.

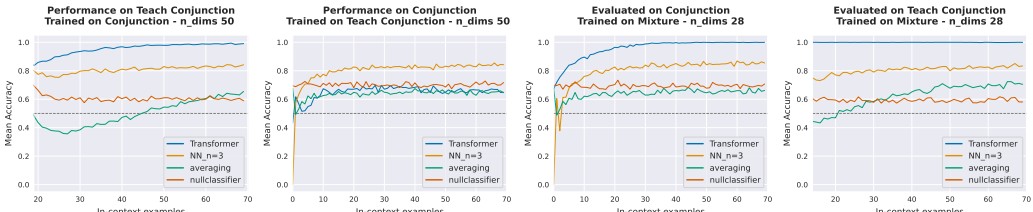

Figure 3: *Left*: depicts the performance of Transformers trained on examples with the vanilla distribution and tested on in-context examples with Teaching Sequences. *Center-left*: Transformers trained with teaching sequences and tested on in-context examples without teaching sequences. *Center-right* and *right*: depict the performance of Transformers trained on a mixture of Conjunction and Teach Conjunction and tested on the individual tasks. See Section 4 for more details.

# 5  INVESTIGATIONS WITH PRETRAINED MODELS

**Premise.** While several works have investigated how well Transformers can be trained to learn various kinds of functions in-context, it is unclear how these findings relate to LLMs, which was the primary motivation behind this line of work. This raises a very natural and intriguing question: *Can pretrained language models predict accurately by implementing any kind of learning algorithm when provided with prompts of the form* $(\mathbf{x}_1, \mathbf{y}_1, \ldots, \mathbf{x}_k)$?

While prior works focused on learning real-valued function classes, evaluating LLMs on those tasks is not straightforward since LLMs receive and produce discrete values. Our setup involves discrete Boolean functions, which enables us to analyze the performance of LLMs in applying learning algorithms for solving these tasks. Additionally, as we are picking the target function randomly from an exponentially large set, we can be essentially sure that the learning is happening from in-context examples only, allowing us to remove confounding factors such as memorization during pre-training.

**Setup.** Recall that, in our experiments in the previous sections, when a model is provided with a sequence of input-output pairs $(\mathbf{x}_1, \mathbf{y}_1, \ldots, \mathbf{x}_m, \mathbf{y}_m)$, each input $\mathbf{x}_i$ is a single token. Additionally, LLMs do not have embeddings for each of the Boolean inputs $\{0,1\}^n$ as opposed to the linear map $\mathbf{W} : \{0,1\}^n \to \mathbb{R}^{\text{width}}$ learned by the models in the ICL setup of previous experiments. To address that, we experiment with pretrained models in two different settings, one which is closer to the ICL setup in previous experiments and another which is very close to the practical usage of the LLMs.

**(a) Frozen GPT.** In the first setting, we take a GPT-2 model and add learnable input and output layers at the beginning and end of the Transformer architecture. The weights of the GPT-2 model are frozen while the embedding matrix is replaced by a learnable function $I : \{0,1\}^n \to \mathbb{R}^{\text{width}}$ which is an FFN with one hidden layer. Similarly, the output layer is replaced by an FFN $F : \mathbb{R}^{\text{width}} \to \{0,1\}$. During experiments, we train the input and output layers to minimize the objective in Eq. 1 while keeping the weights of the pre-trained GPT-2 frozen. We compare it with the performance of a randomly initialized Transformer model (same architecture as GPT-2) undergoing the same training process with learnable input-output mappings and frozen Transformer weights. The goal of this setup is to understand whether large-scale text pre-training imparts some ability to the Transformer model that enables it to implement learning algorithms in-context.

**(b) Direct Evaluation.** We directly evaluate several LLMs on the task of learning Boolean functions. In this setup, each input example $\mathbf{x}_i \in \{0,1\}^n$ is provided as a sequence of tokens $x_1, \ldots, x_n$ where $x_i \in \{0,1\}$. None of the model's parameters are modified and the original embedding matrices are used to represent the tokens $0$ and $1$. We evaluate the models by providing them with a simple prompt containing two parts: the first part is a simple instruction asking the model to act as a learning algorithm and predict the label corresponding to the final input, and the second part consists of a series of input sequences and their corresponding labels followed by the test input sequence (see Figure 21). We evaluate the predictions of the model at prompts of different lengths (i.e., different numbers of in-context examples). To estimate the accuracy for a particular number of in-context examples, we repeat the experiment 100 times (i.e., with 100 different functions of the same class). We experiment with state-of-the-art LLMs GPT-4 (OpenAI, 2023), GPT-3.5-Turbo (Brown et al., 2020; Ouyang et al., 2022) as well as the open-source model LLaMA-2-70B (Touvron et al., 2023).

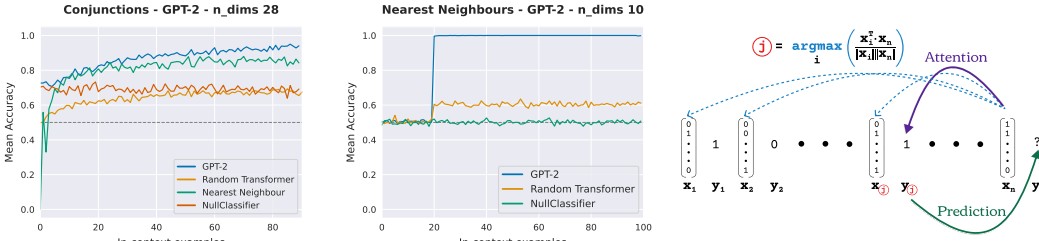

Figure 4: Experiments with frozen GPT: *left*: Performance of frozen GPT-2 on the Conjunction task, *center*: Frozen GPT-2 model learns to perfectly implement the nearest neighbors algorithm, *right*: Illustration of attention patterns in GPT-2 while solving the Nearest Neighbor task.

## 5.1 RESULTS: LLMS PERFORM AT LEAST AS WELL AS NEAREST NEIGHBORS

**Results with Frozen GPT.** We evaluate the performance of a frozen GPT-2 model on tasks such as Conjunctions and Disjunctions. Figure 4 depicts the performance on Conjunctions (the behavior on Disjunctions is almost identical). We find that the GPT-2 model performs relatively better than the nearest neighbor baseline and much better than a randomly initialized model on these tasks. However, it still does not come close to achieving the near-perfect accuracy obtained by fully trainable Transformer networks as in Section 3.

**GPT-2 can implement Nearest Neighbor.** Given the observation that the performance of GPT-2 was close to the nearest neighbor algorithm, we examined if a frozen GPT-2 model can implement the nearest neighbor (NN) algorithm. We designed the NN task to test this hypothesis. In this task, each prompt contains $100$ points where the first $20$ points are labeled $0$ or $1$ uniformly at random and the subsequent $80$ points are labeled according to the nearest neighbor algorithm. We then evaluate the GPT-2 model in the frozen-GPT setup described earlier and find that it can achieve near-perfect accuracy on the $80$ points labeled with the nearest-neighbor algorithm (see Figure 4 center). Moreover, upon analyzing the attention heads we found heads which closely implemented nearest neighbors — for an input $\mathbf{x}_i$, the attention head attended over $\mathbf{y}_j$ where $\mathbf{x}_j$ is the nearest neighbor of $\mathbf{x}_i$ among $\mathbf{x}_1, \ldots, \mathbf{x}_{i-1}$ (illustrated in Figure 4 right). Further details about these experiments are provided in Appendix H. These heads are reminiscent of induction heads (Olsson et al., 2022) where instead of matching prefixes, they can find the nearest neighbor over the input space. Since the weights of the Transformers are frozen and only the input embedding and output layers are updated during the training process, the mechanisms to solve tasks such as NN and Conjunctions must have been learned during pretraining on language data. Moreover, a Transformer of the same size with randomly initialized weights is unable to perform much better than chance-level accuracy.

**Results with Direct Evaluation.** The performances of all LLMs for the Conjunction and Majority tasks with varying number of dimensions are provided in Figure 5. It is clear that all models are able to perform as well as or better than the nearest neighbor baseline when the number of dimensions is up to 7. Note that in this setup, the LLM is essentially unaware of the task (such as Conjunction) when it is provided directly with the prompt at inference time. Even with $n = 7$, there are $2^{128}$ Boolean functions, and so the in-context learning problem remains challenging, and the observed performance of these models is impressive. It is perhaps surprising to see that the open-source LLaMA-2 model performs quite similar to GPT-3.5-Turbo. It is also particularly interesting to see the strong performance of GPT-4; apart from outperforming all other models, it also slightly outperforms the nearest neighbor baseline even in the 15-dimensional case.

## 6 RELATED WORK

**In-Context learning in Stylized settings.** Understanding the in-context learning (ICL) phenomenon has attracted significant interest in recent years. Since the ICL phenomenon was demonstrated in Brown et al. (2020), numerous works have investigated the practical capabilities and limitations of LLMs to perform ICL (see Dong et al. (2022) for a survey). Since it is infeasible to precisely define real-world data and tasks, several works have studied it in stylized well-defined settings (Chan et al., 2022; Hahn & Goyal, 2023; Xie et al., 2021). Garg et al. (2022) presented a meta-learning-like

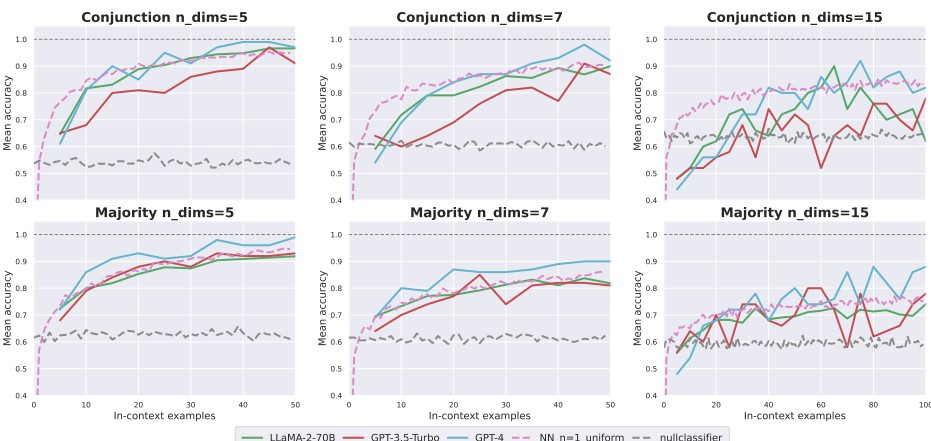

Figure 5: Results with the direct evaluation of LLMs at inference time. The *top* row shows the performance across varying dimensions for the Conjunction task while the *bottom* row shows the performance for the Majority task.

framework and demonstrated that Transformers can match gradient-based learning algorithms for regression tasks. Subsequent works (Dai et al., 2022; Akyürek et al., 2023; von Oswald et al., 2022) demonstrated that Transformers were expressive enough to represent gradient-descent and empirically verified that the learned weights indeed computed gradient-based algorithms. Li et al. (2023); Bai et al. (2023) studied the sample complexity of in-context learning regression and other real-valued functions with Transformers. More recently, Ahuja et al. (2023); Bai et al. (2023) explored (among other things) the efficacy of Transformers in learning mixtures of tasks in-context and showed that they could adaptively select appropriate algorithms for a given sequence of inputs and labels. Our work takes a step forward in this direction by demonstrating that (a) some of these observed phenomena are not confined to attention-based architectures and (b) that neural networks can learn to implement multiple distinct learning algorithms for ICL of a single task depending on the in-context examples. Moreover, our results help bridge the gap between the stylized setup used in prior works and in-context learning of actual pretrained models.

**Transformers and Sequence Models.** The analysis of the capabilities and limitations of recurrent architectures dates back to a few decades ago (Kolen & Kremer, 2001). Given the recent success of Transformers, several works have sought to investigate their theoretical expressiveness (Pérez et al., 2019; Merrill et al., 2022; Chiang & Cholak, 2022; Hahn, 2020; Yun et al., 2020; Liu et al., 2022) as well as their empirical capabilities (Bhattamishra et al., 2023; Ebrahimi et al., 2020) and limitations (Bhattamishra et al., 2020; Chiang & Cholak, 2022). Delétang et al. (2022) conduct a comprehensive study of the performance of various sequence models such as Transformers and RNNs on formal language tasks. While most of these prior works focus on classification or related tasks, our work complements these as we conduct a comprehensive study on in-context learning tasks.

# 7 CONCLUSION

In this work, we highlighted certain limitations of Transformers in in-context learning and their ability to leverage more informative examples such as teaching sequences for learning in a more sample-efficient manner. Additionally, we showed that pretrained models also learn mechanisms to solve tasks in the stylized ICL setting. Further, we show that LLMs are also capable of performing as well as certain baseline learning algorithms in a more practical setting. Our results raise several interesting questions for future work: (a) Can LLMs leverage more informative examples to sample-efficiently learn practical tasks in-context? (b) What aspects of the pretraining process of LLMs lead to their ability to implement learning algorithms since they are not primarily trained in the meta-learning-like framework? (c) What are the theoretical limitations behind the difficulty of learning Parities in the in-context learning setting? (d) What are the mechanisms that various architectures (attention, recurrence, and convolution) use to learn tasks using different operations?

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

## A  ROADMAP

The appendix is organized as follows.

- In Section B, we discuss clarifications related to some natural questions related to our results.
- In Section C, we provide detailed descriptions of the tasks, the distribution of functions and inputs used in our experiments, and details of the baselines. Additionally, the detailed plots regarding some results reported in Section 3 are provided in this section.
- In Section D, we discuss results with some additional experiments exploring the robustness of the models to out-of-distribution tasks and to noisy labels as well as the sample complexity of models while learning tasks such as Conjunctions. We also discuss some factors that could determine the difficulty of tasks for sequence models.
- In Section E, we compare the performance of Transformers with known algorithms for Conjunction and study their representational complexity.
- In Section F, we discuss the difficulty of learning the Parity task in more detail.
- In Section G, we provide the details of the teaching sequences used for the experiments in Section 4.
- In Section H, we provide some additional details related to the experiments with LLMs in Section 5 and also discuss some additional results.
- In Section I, we discuss the details of the implementation and hyperparameters used during the experiments.
- In Section J, we discuss some additional related work.

## B  CLARIFICATIONS

**(1)** *Given that LLMs are trained on trillions of tokens, how can we be certain that LLMs such as GPT-4 are applying learning algorithms and not imitating any tasks seen during training?* In prior works where in-context learning is evaluated with natural language tasks such as sentiment classification, NLI, or some variations of them, LLMs are likely to have seen such tasks during training and can use in-context demonstrations for recognizing the target function learned during pretraining before making predictions (Min et al., 2022b). However, in our case where the model is provided with a sequence of Boolean inputs and labels (in Section 5), the total number of possible target functions that can label the inputs is over $10^{30}$ even for dimension 7 so it is virtually impossible for a model to have seen sequences of examples labeled with such a large amount of functions. Hence, in order to predict accurately, the model has to rely on the examples provided in-context as a learning algorithm.

**(2)** *When you say LLMs apply learning algorithms, is the claim that they implement gradient-based or nearest neighbor algorithm?* No. We do not know how LLMs are learning based on the in-context examples provided. Our results suggest that they are competitive with the nearest neighbor (NN) algorithm but we do not have any evidence to claim that they are applying NN, gradient-descent, or any specific algorithm for that matter for learning tasks like Conjunctions.

**(3)** *Are Transformers superior to other architectures in in-context learning in the sense of implementing learning algorithms?* Not necessarily. Our results across various classes of Boolean functions and linear regression suggest that attention-free architectures match Transformers on most tasks and are slightly worse on a few tasks. The difference in the nearest neighbor task is particularly interesting because our results in Section 5 suggest that Transformers learn such mechanism during their pretraining process. At the same time, one could argue that the task is more suited to attention-based models. Given the similarity in performance between Transformers and Hyena on other tasks, it would not be surprising if there are learning problems that are more suited to long convolutional models where Hyena could perform better.

**(4)** *Why are the dimension of inputs (n_dims) and number of examples (n_points) different for different tasks in Section 3?* Parameters such as the n_dims and n_points are chosen so as to ensure feasibility of the learning task to some degree based on known results about the learnability of the problem. For instance, for tasks such as Conjunctions and Disjunctions, since there exists known PAC learning algorithms (Valiant, 1984) for solving the task with $O(n)$ examples, we can set the n_dims as 28

and expect models to achieve near-perfect accuracy within 70 examples. However, for tasks such as Parities, since gradient-based algorithms need $n^{\Omega(k)}$ steps to learn them (Kearns, 1998), we consider the PARITY-$(10, 2)$ task in case Transformers can learn using a gradient-based algorithm similar to FFNs with gradient descent. On the other hand, since Parities can be learned within $O(n)$ examples with Gaussian elimination algorithm (Fischer & Simon, 1992), we consider the PARITY-$(20, 3)$ and PARITY-20 task where models will have to find a non-gradient-based algorithm to solve them.

**Limitations.** A caveat with empirical studies such as this is that the results depend on the hyperparameters and other aspects of the experimental setup. While we have tried to be as thorough as possible with hyperparameter tuning, there is always a chance that the results or behaviour could differ for some hyperparameter.

## C  ADDITIONAL DETAILS OF EXPERIMENTS

### C.1  DESCRIPTIONS OF TASKS

Boolean functions have been widely studied in the field of theoretical computer science (O'Donnell, 2021). Given that it is extensively studied in the field of learning theory, it is well-suited for our task, since the discrete nature of the domain allows us to also test actual LLMs apart from neural networks trained from scratch. Additionally, the learnability and the precise properties of each function class are well-understood which aids in experimental design choices (such as the number of in-context examples necessary for learning a concept class) and in drawing inferences from the results.

For better readability, we redefine the function classes that are already described in the main paper.

**Tasks 1-2** *Conjunctions and Disjunctions.* For the input domain $X_n = \{0,1\}^n$, any input $\mathbf{x} \in X_n$ denotes a possible assignment to $n$ Boolean variables $x_1, \ldots, x_n$. A literal is either the variable $x_i$ or its negation $\bar{x}_i$. A conjunction is simply an *and* ($\wedge$) of some subset of the $2n$ literals. For examples, for $n = 10$, one possible conjunction could be $x_2 \wedge \bar{x}_6 \wedge x_7$ which outputs 1 only when $x_2, x_7$ are 1 and $x_6$ is 0. The class of Conjunctions contains all such conjunctions over $\{0,1\}^n$. The class of Disjunctions is defined similarly with the *or* ($\vee$) operation ($x_2 \vee \bar{x}_6 \vee x_7$).

**Task 3** *Sparse Disjunctions* is a specific case of Disjunctions where the function over $n$ variables depends on at most $k$ variables where $k \ll n$. In other words, the function class contains all Disjunctions with at most $k$ variables. In our experiments, we set $k = 3$. For functions over $n$ variables, the total number of 3-sparse disjunctions is $\binom{n}{3}3^3$. Hence, in our experiments, we set a higher value for the $n$ (n_dims$= 50$) parameter so that the number of functions in the class is at least a million.

**Task 4-5** *DNFs and CNFs.* A Boolean function is in the class DNF if it is a disjunction of one or more conjunctions. Formally, for any $\mathbf{x} \in \{0,1\}^n$, a function $f$ belongs to the class of DNFs, if it can be expressed in the form $f(\mathbf{x}) = (z_{1,1} \wedge z_{1,2} \wedge \ldots \wedge z_{1,k_1}) \vee \ldots \vee (z_{m,1} \wedge z_{m,2} \wedge \ldots \wedge z_{m,k_m})$ where $z_{i,j}$ are literals which are either the variables or their negation. Similarly, CNFs contain functions that are a conjunction of one or more disjunctions. While only the simplest of classification rules may be defined using conjunctions/disjunctions, a richer family becomes available when using DNF/CNF representations. Since DNFs/CNFs are conjectured to be computationally hard to learn, we consider 3-term DNFs and 3-clause CNFs which are known to be efficiently learnable.

**Task 6** *Sparse Majority.* A function in the class sparse majority is parameterized by a subset of variables and outputs the majority value of those variables. For example, if the relevant variables are $x_2, x_4, x_5$ and $x_7$, then the output for any input $\mathbf{x} \in \{0,1\}^n$ will be 1 if more than 2 variables have value 1 and will be 0 otherwise. Since each function is uniquely identified by a subset of variable $S \subseteq [n]$, the total number of sparse majority functions is $2^n$.

**Task 7** *0-1 Threshold Functions* are functions of the form $\text{sign}(\sum_{i=1}^{n} w_i x_i - b)$ where $w_1, \ldots, w_n \in \{-1, 0, 1\}$ and $b \in \{-k, \ldots, k-1, k\}$. Instead of the usual $X_n = \{0,1\}^n$, the input space can be seen over $\{-1, +1\}^n$ which makes it easier to define the functions. In our experiments, we set $k = 3$ and $n = 28$. Note that, Conjunctions and Disjunctions can be seen as special cases of Integer threshold functions where $b$ is $\|w\|_1 - 1$ or $-(\|w\|_1 - 1)$. See that the total set of functions in the class of Integer threshold functions is $\Omega(3^n)$.

**Task 8** *Integer Halfspace* contains functions of the form $\text{sign}(\sum_{i=1}^{n} w_i x_i - 0.5)$ where $w_1, \ldots, w_n \in \{-k, \ldots, k-1, k\}$ and $x \in \{-1, +1\}^n$. The total set of functions in the class is $\Omega(3^n)$.

**Tasks 9-11** *Parity* is one of the most fundamental classes of Boolean functions; widely studied in learning theory and cryptography. A Parity function computes the XOR ($\oplus$) of some subset of the $n$ variables. For example, a Parity function could be $x_2 \oplus x_4 \oplus x_5$. Each Parity function outputs 1 when an odd number of the variables are assigned the value 1 and outputs 0 otherwise. The class PARITY-$n$ contains all $2^n$ possible Parity functions over $\{0, 1\}^n$. We denote the class of sparse parities with PARITY-$(n, k)$ which contain Parity functions with exactly $k$ relevant variables (or bits) defined over $\{0, 1\}^n$.

**Task 12** The Nearest Neighbour task is intended to evaluate the ability of architectures to learn to implement the nearest neighbour algorithm. Unlike, other tasks, it doesn't involve a class of functions. For this task, each example consists of 100 points $x_1, \ldots, x_1 00 \in \{0, 1\}^n$ where the first $k$ points are labelled 0 or 1 uniformly at random. The labels for the next $100 - k$ points are determined by their nearest neighbour in the preceding input points. More specifically, for any point $x_j$ where $j \in \{k+1, \ldots, 100\}$, let $p = \arg\max_{i \in \{1, \ldots, j-1\}} \frac{x_i^T x_j}{\|x_i\| \|x_j\|}$. Then the label $y_j$ corresponding to $x_j$ is $y_p$.

**Size of Hypothesis set.** Unlike regression problems considered in earlier works, the number of functions in these Boolean function classes is finite. However, see that, for inputs over $\{0, 1\}^n$, the total number of possible Conjunctions (or Disjunctions) is $3^n$. Hence, even for $n = 30$, the function class has over 100 trillion functions. Moreover, even for a prompt with $m = 50$ examples, the total number of permutations is 50! for one set of inputs and a single target function. This implies that the likelihood of encountering the same training example twice is extremely low and that it is (almost) impossible for a practical-sized model to memorize all examples during training unless $n$ and $m$ are quite small.

## C.2 DETAILS OF INPUT AND FUNCTION DISTRIBUTION

For the majority of the tasks, the inputs are sampled uniformly at random over $\{0, 1\}^n$. For tasks such as Conjunctions, Disjunctions, and 3-CNF, a uniform distribution for the inputs over $\{0, 1\}^n$ will lead to most examples being either positive or negatively labelled. In that case, even the Null classifier which always produces the same output can achieve over 99% accuracy. While Transformers trained over uniformly distributed inputs for these tasks achieve near-perfect accuracy, it is in some sense trivial since they do not necessarily need to learn any algorithms and can achieve near-perfect accuracy by learning a constant function. Hence, for such tasks, we modify the input distribution such that at least 30% of the input points in the prompt have a positive (or negative) label to make it more well-balanced. In the modified distributions, we first sample $k$ points for the prompt uniformly at random and then pick 30% of the input points. For these points, we change a subset of bits to 0 or 1 so that they will labelled positively (for Conjunction) or negatively (Disjunction) according to the target function. Note that, this doesn't change the proportion of 0s and 1s in the modified inputs since the probability of a literal being positive or negative (complement) in the Conjunction is the same.

**Distribution over Functions.** For tasks such as Sparse Disjunctions and Parities, the functions are sampled uniformly from the set of all such functions. For both 0-1 Threshold functions and Integer halfspace, the $w_i$s are sampled uniformly from their respective range and for 0-1 Threshold functions, the $b$ parameter is sampled uniformly from $\{-k, \ldots, k-1, k\}$. For tasks such as Conjunctions and Disjunctions which are either AND ($\wedge$) or OR ($\vee$) of some of the $2n$ literals, the function is sampled such that each literal $x_i$ or its complement $\bar{x}_i$ has a probability of $p$ in being in the sampled function. In other words, for each $i \in [n]$, the literal $x_i$ has $p/2$ probability of being sampled, the literal $\bar{x}_i$ has $p/2$ probability and with $1 - p$ probability the literal or it's complement is not included. In our main experiments, we set $p = 30\%$ and also investigate the robustness of the trained model when evaluated on other distributions (other values of $p$) in Appendix D.1. Similarly, for the Sparse Majority class, the probability of each literal being in the sampled function is $p$ (there is no complement). For sampling each 3-DNF (or 3-CNF), we first sample three conjunctions (or disjunctions) with $p = 20\%$ and then take an AND (or OR) of those clauses.

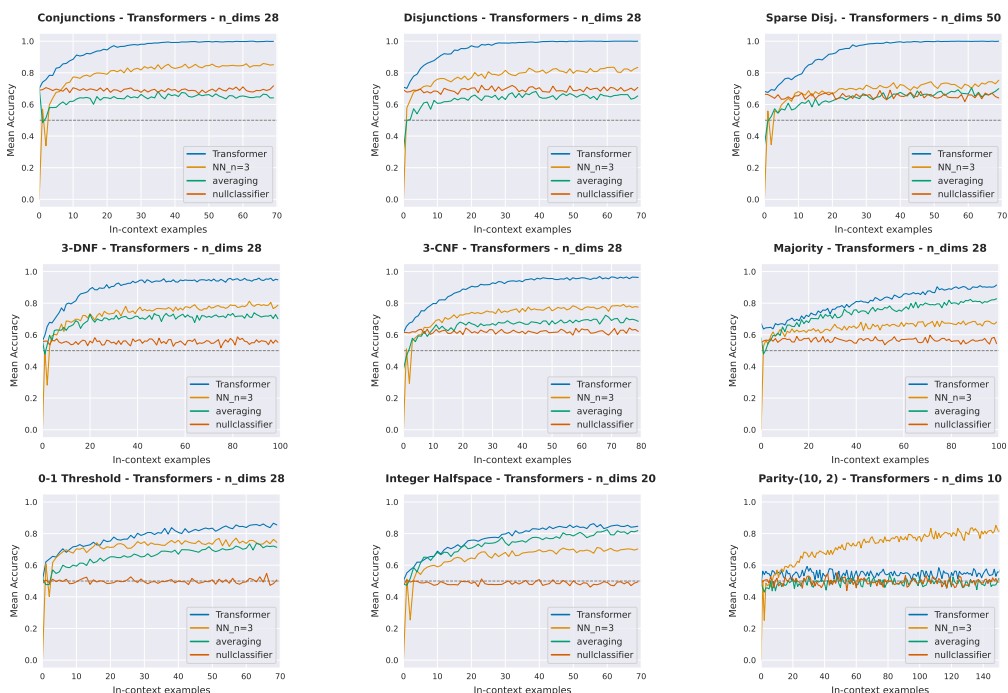

Figure 6: Performance of Transformers on in-context learning various classes of Boolean functions. The plots represent the mean accuracy of Transformers as a function of the number of in-context examples provided to the model. On certain classes of Boolean functions such as Conjunctions and Disjunctions, Transformers perform in a near-optimal manner. On tasks such as 0-1 Threshold, Majority and Integer Halfspaces, they perform better than baselines but do not achieve near-perfect accuracy even after observing 70 examples. On the other hand, on Parities Transformers do not perform better than chance-level accuracy. Refer to Section 3 for details.

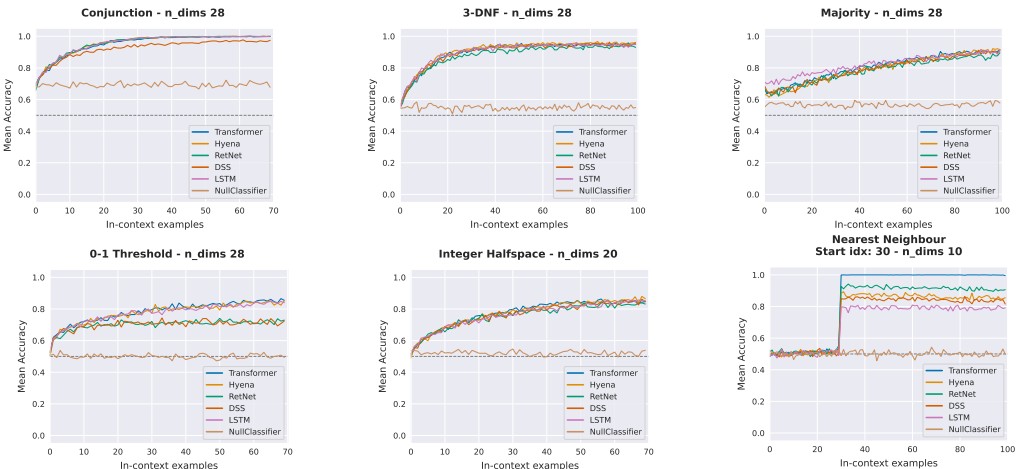

Figure 7: Performance of different architectures on 6 representative tasks.

## C.3 DETAILS OF BASELINES

Below, we describe how each baseline model $M$ predicts the label when provided with a prompt $P_k = (\mathbf{x}_1, \mathbf{y}_1, \ldots, \mathbf{x}_{k-1}, \mathbf{y}_{k-1}, \mathbf{x}_k)$.

**Null classifier.** This classifier always produces the same output, i.e. $M(P_k) = 0$, irrespective of $P_k$.

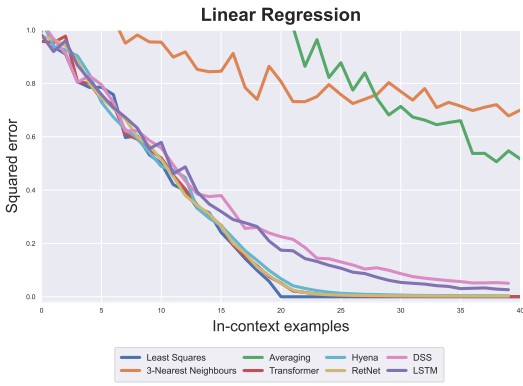

Figure 8: Performance of various architectures on the Linear Regression task from Garg et al. (2022).

**Averaging estimator.** This is an incremental learning classifier that captures the average relationship between input vectors and output labels for previously seen points. It predicts $M(P_k) = \hat{\mathbf{w}}^T \mathbf{x}_k$. The weights $\hat{\mathbf{w}}$, are computed as the average of the product of input vectors and their corresponding outputs over all previous points, i.e., $\hat{\mathbf{w}} = \frac{1}{k-1} \sum_{i=1}^{k-1} \mathbf{x}_i \mathbf{y}_i$.

**Nearest neighbors (NN).** This classifier aims to predict the output label based on the labels of the $n$ most similar input vectors in the previously seen points. It predicts $M(P_k) = \frac{1}{n} \sum_{i \in S} \mathbf{y}_i$. Here, $S$ is the set of indices of the $n$ nearest neighbors of $\mathbf{x}_k$ among $\mathbf{x}_1$ to $\mathbf{x}_{k-1}$. For $k - 1 < n$, we average over all the $\mathbf{y}_i$s from 1 to $k - 1$, and for $k = 1$, we set $M(P_k) = 0$.

**Feedforward Networks (FFN).** We train a 1-hidden layer ReLU FFN on the same number of examples as provided to sequence models in Table 1. We tune the model across width $\in \{100, 500, 1000\}$, optimizer { 'adam', 'sgd'}, and learning rates $\in \{0.01, 0.005, 0.001, 0.0001\}$. We report the accuracy of the best model based on 1280 examples after being trained on $m - 1$ examples where $m$ is given in Table 1 for each task.

### C.4 Additional Plots for In-context learning Boolean functions

The accuracy curve of Transformers and baselines across all in-context examples is provided in Figure 6. The performances on Conjunctions and Disjunctions are almost identical in all the experiments we have conducted. Similarly, the performance on 3-CNF and 3-DNF as well as among Parity tasks are almost identical. The accuracy curves of all 5 architectures on representative tasks are provided in Figure 7.

## D Additional Experiments on Boolean Functions

### D.1 Robustness of Models

In this section, we explore the robustness of Transformers to out-of-distribution (OOD) inputs and functions as well as their robustness to noisy examples during training.

**OOD functions.** Transformers seemingly perform in a near-optimal manner for tasks such as Conjunctions and Disjunctions. First, we evaluate the performance of Transformers trained on the Conjunctions task on functions from a different distribution than the ones seen during training. As described in Appendix C, while training on the Conjunctions task, the functions are sampled in such a way that for any $i \in [n]$, the probability of the literal $z_i$ or $\bar{z}_i$ being in the sampled Conjunctions is 30%. Hence, the expected number of literals in the Conjunctions seen during training is $n/3$. We evaluated Transformers trained on such Conjunctions on examples where the functions are sampled in such a way that the expected number of literals is $2n/3$ and the probability of any literal being in the sampled functions is twice as compared to the ones seen during training. We find that Transformers perform effectively on such examples as well (see Figure 9 Right). However, we find that the

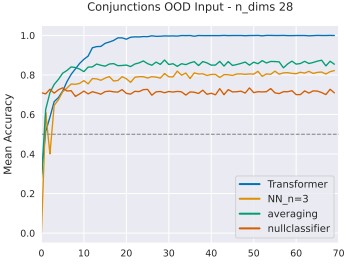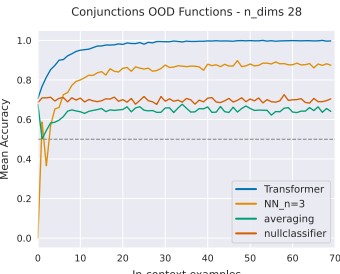

Figure 9: Performance of Transformers on Conjunctions with out-of-distribution inputs (left) and functions (right). Refer to Section D.1 for more details.

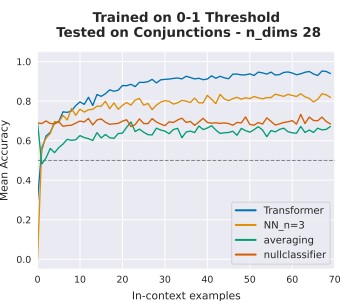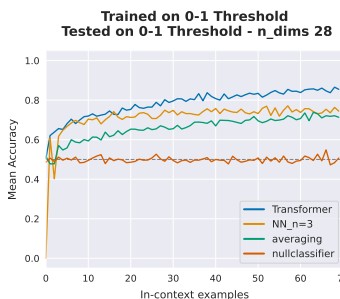

Figure 10: Comparison between the performance of a Transformer trained to learn the 0-1 Threshold function on the Conjunction task (left) and the original 0-1 Threshold task. Refer to Section D.1 for more details.

performance degrades for extreme changes in distribution where the expected number of literals in the functions is above $80\%$.

**Trained on Threshold and Tested on Conjunctions.** We also evaluated the performance of Transformers trained on 0-1 Threshold functions on the Conjunctions task. As describe in Appendix C, 0-1 Threshold Functions are functions of the form $\mathrm{sign}(\sum_{i=1}^{n} w_i x_i - b)$ where $w_1, \ldots, w_n \in \{0,1\}$ and $b \in \{-k, \ldots, k-1, k\}$. In some sense, Conjunctions are simpler threshold functions where the function is of the form $\mathrm{sign}(\sum_{i=1}^{n} w_i x_i - (\|w\|_1 - 1))$. At the same time, note that the 0-1 Threshold functions task in our setting will not have functions which are exactly Conjunctions since $k = 3$ in our experiments and $\mathbb{E}[\|w\|_1] = n/3$ where $n$ is set to 28 for experiments with Threshold functions.

We conduct an experiment where we train a Transformer on the 0-1 Threshold function task (with $n = 28$) and evaluate the model on the Conjunctions task. Perhaps surprisingly, we find that the model performs better on the Conjunctions task (achieving over $90\%$ at the end) in comparison to the in-distribution Threshold function task (See Figure 10). It is interesting to see that the algorithm learned during training generalises to higher values of $b$ in $\mathrm{sign}(\sum_{i=1}^{n} w_i x_i - b)$. These observations could perhaps be attributed to the fact that Conjunctions are one of the simplest Threshold functions and could be easier since the value of $b$ is dependent on $w$.

**Robustness to noise.** On tasks such as Conjunctions and Disjunctions where the models learn to perform in-context learning effectively, we explore whether they are robust to noisy labels during the training process. During training, for every example $(\mathbf{x}_1, \mathbf{y}_1, \ldots, \mathbf{x}_m, \mathbf{y}_m)$, we flip $10\%$ of the labels $\mathbf{y}_i$s. Hence, the examples seen during training are not perfectly labeled by a Conjunction (or Disjunction). During the evaluation, we provide clean inputs (without noise) to test the performance of the models. We find that on these tasks, all architectures are robust to noise in the training process (see Figure 11).

**Length Generalization.** We explore the effectiveness of Transformers to generalize to a larger number of examples (in some sense length) in comparison to those seen during training. We consider Conjunctions and DNFs where they perform reasonably well in the in-distribution setting. For Conjunctions, we train the model on prompts with 60 examples and test their performance on prompts

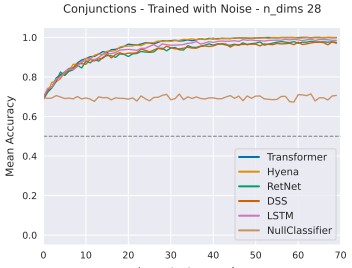 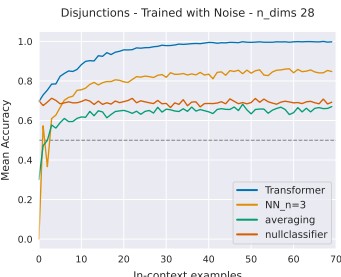

Figure 11: Left: Performance of various architectures trained with noisy examples on the Conjunctions task. Right: Performance of Transformers trained with noisy examples on the Disjunctions task and performance of baselines. Refer to Section D.1 for more details.

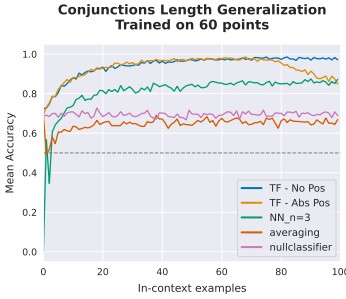 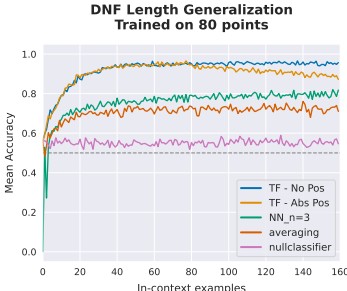

Figure 12: Performance of Transformers on prompts with a larger number of examples than seen during training. Left: Trained on Conjunctions with 60 examples and tested on 100. Right: Trained on DNFs with 80 examples and tested on 160. Transformers with no positional encodings and just masking generalize well whereas those with absolute encodings struggle on higher lengths. Refer to Section D.1 for more details.

with 100 examples. Similarly, for DNFs, we train the model on prompts with 80 examples and test their performance on prompts with 160 (twice) examples. We compare two types of positional encodings: (a) absolute encodings and (b) no positional encodings with just causal masking. Figure 12 depicts the performance of Transformers on these tasks. Transformers with no positional encodings and just masking seem to generalize well whereas those with absolute positional encoding seem to struggle on higher lengths.

## D.2 WHAT DETERMINES THE DIFFICULTY OF IN-CONTEXT LEARNING?

A natural question that arises from our results is: what properties of a task determine the difficulty of in-context learning for models such as Transformers and other architectures? First, it is interesting to observe that capacity measures such as VC dimension or the size of a function class do not seem to correlate well with the performance of the models (See Table 2). While Transformers are able to perform well on Conjunctions which have VC dimension $O(n)$ and sparse Disjunctions which have VC dimension $O(k \log n)$, they fail on Parities and sparse Parities with similar VC dimensions. VC dimension is one way to measure the complexity of a class of functions which relates to the worst-case sample complexity required to learn functions from the class and it is related to several other measures which compute the capacity of the class in some manner.

Note that, measures such as VC dimension are not dependent on either the distribution of inputs or the functions. We explore some other measures in order to understand the properties that determine the difficulty of in-context learning. One measure which we found to correlate well with the performance of Transformers is the pairwise correlation between the functions in a class. For Boolean functions of the form $f : \{0, 1\}^n \rightarrow \{\pm 1\}$, the pairwise correlation of a class of functions $P(\mathcal{F})$ is defined as,

$$P(\mathcal{F}) = \left| \mathbb{E}_{f_1, f_2} [\mathbb{E}_{x \sim U} [f_1(x) f_2(x)]] \right|. \tag{2}$$

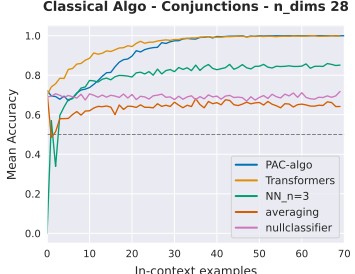 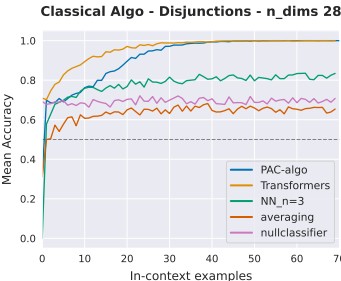

Figure 13: Comparison between the performance of classical PAC-learning algorithms for Conjunctions and Disjunctions and the performance of Transformers. See Section E for the description of the algorithm and other details.

The measure $P(\mathcal{F})$ computes the magnitude of the expected correlation between pairs of functions sampled over $D_{\mathcal{F}}$. To estimate $P(\mathcal{F})$ for the tasks considered in the paper, we sample 1k functions according to the distribution used for training and sample 10k inputs from the uniform distribution over $\{0, 1\}^n$. The estimated pairwise correlation of all function classes is provided in Table 2. See that the pairwise correlation between any two Parity functions is 0. We observe that the value $P(\mathcal{F})$ is high for classes in which models perform well and decreases across categories with Parities having the minimum pairwise correlation.

While the pairwise correlation among functions has a reasonably strong correlation with the degradation of the performance of the models, it still has some weaknesses. As compared to properties of function classes such as VC dimension, it takes the distribution over functions into account. However, the input distribution is uniform over $\{0, 1\}^n$ while computing $P(\mathcal{F})$ whereas it is not uniform for some of the function classes during training. Having a clearer understanding of the factors that determine the difficulty of in-context learning is an interesting open problem.

Table 2: Pairwise correlations of different functions classes computed according to Eq. 2. See Section D.2 for more details.

| Tasks | Conjunctions | Disjunctions | Sparse Disjunctions | 3-CNF | 3-DNF | Majority | Integer-Threshold | Integer Halfspace | Parity-(10, 2) | Parity-20 |
|---|---|---|---|---|---|---|---|---|---|---|
| VC Dimension | $O(n)$ | $O(n)$ | $O(k \log n)$ | $O(n)$ | $O(n)$ | $O(n)$ | $O(n)$ | $O(n)$ | $O(k \log n)$ | $O(n)$ |
| Pairwise Correlation | $95.2 \pm 0.07$ | $95.3 \pm 0.08$ | $77.1 \pm 0.01$ | $51.86 \pm 0.29$ | $50.7 \pm 0.26$ | $20.7 \pm 0.1$ | $0.6 \pm 0.21$ | $0.1 \pm 0.14$ | $0.08 \pm 0.03$ | $0.01 \pm 0.03$ |

### D.3 EXPLORING SAMPLE COMPLEXITY

In this section, we explore the performance of Transformers when the number of examples during training is fixed. Recall that each training (or test) example $X = (\mathbf{x}_1, \mathbf{y}_1, \ldots, \mathbf{x}_m, \mathbf{y}_m)$ is created by sampling $m$ input points from the distribution $D_X$ over inputs and sampling one target function from the distribution $D_{\mathcal{F}}$ over functions. For the experiments in Section 3, a fresh set of input points and functions are drawn during each training iteration.

For example, while training Transformers for 20k steps for the task Conjunctions with batch size 64, the model essentially observes about 1.3M functions and input sequences. However, note that the likelihood of observing the same function during evaluation is extremely low since for 28 dimensions, the total number of Conjunctions is over 22 trillion.

An interesting question is to understand how many examples Transformers need to generalize well in the in-context learning setting. Moreover, what is the effect of increasing the capacity of the model when the number of examples remains fixed? We explore these questions in this section.

We consider Conjunctions where Transformers are effective in the vanilla setting. During training, we first sample a fixed number of examples $N$ of the form $(\mathbf{x}_1, \mathbf{y}_1, \ldots, \mathbf{x}_m, \mathbf{y}_m)$ by sampling $N$ functions from the distribution $D_{\mathcal{F}}$ and sampling $N$ sequence of $m$ input points from the distribution $D_X$. All training iterations over the $N$ examples and during evaluation we sample fresh examples to estimate the accuracy of the model.

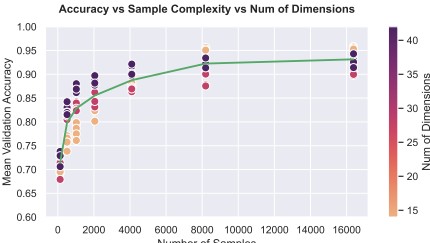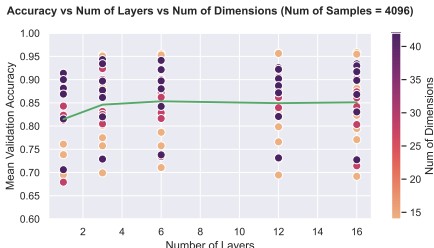

Figure 14: Performance of Transformers on learning to learn Conjunctions with a finite number of examples. The figures show the change in performance across various axes such as depth, number of examples, and number of dimensions in the input. Refer to Section D.3 for more details.

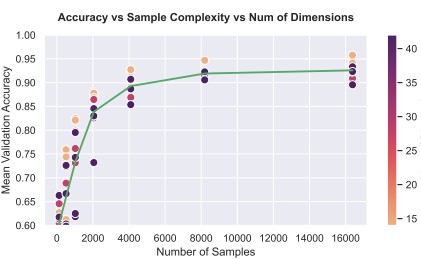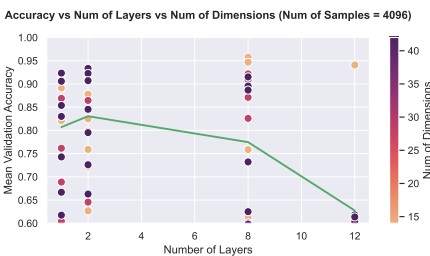

Figure 15: Performance of LSTMs on learning to learn Conjunctions with a finite number of examples. The figures show the change in performance across various axes such as depth, number of examples, and number of dimensions in the input. Refer to Section D.3 for more details.

We evaluate the model with $N \in \{128, 512, 1024, 2048, 4096, 8192, 16384\}$ examples. We test the models across tasks with dimensions $d \in \{14, 28, 42\}$. We find that neural networks such as Transformers and LSTMs are quite sample-efficient in terms of solving Conjunctions task where they achieve close to perfect accuracy even with 4096 examples. Figure 14 (left) and Figure 15 (left) depict the mean accuracy of Transformers and LSTMs across various number of examples $N$ and task dimensions $d$. Note the mean accuracy during evaluation here is the mean of the accuracy of all $m$ points in an example and not the accuracy on the last point. For instance, if we test on $K = 1280$ examples to estimate the accuracy and each example has $m = 70$ points then the mean accuracy is computed as $\frac{1}{K} \sum_{i=1}^{K} \left( \frac{1}{m} \sum_{k=1}^{m} \mathbb{I}[M(P_k^{(i)}) = f_i(\mathbf{x}_k)] \right)$. The mean accuracy across all in-context examples will always be lower than $100\%$ since the model will make mistakes in the first half of the examples before converging to a near-perfect accuracy for tasks like Conjunctions.

It is also interesting to see that increasing the capacity of the model has a negligible effect on the performance of the model. Figure 14 (right) shows the mean accuracy of Transformers across different depths (number of layers) when trained on 4096 examples.

# E  TRANSFORMERS AS LEARNING ALGORITHMS

As described in Section 2, in this line of work each sequence of examples or prompt $(\mathbf{x}_1, f(\boldsymbol{x}_1), \ldots, \mathbf{x}_m, f(\boldsymbol{x}_m))$ can be seen as a learning problem. In the traditional statistical learning framework, while learning from examples labelled by a particular function $f \in \mathcal{F}$ (Conjunction), the goal of a learning algorithm is to find the target hypothesis function $f \in \mathcal{F}$. In this case, during the meta-learning stage, we are optimizing a Transformer (or other networks) to find a learning algorithm rather than a target function. Hence, in that sense, the target function in this meta-learning-like setup is the optimal learning algorithm for the class of functions. It is natural to wonder how Transformers fare against the optimal learning algorithms for the problems considered in this work and moreover, if they are theoretically capable of representing and learning such learning algorithms. While the optimal learning algorithms for all classes considered here may not be known, there are some known algorithms for learning classes such as Conjunctions and Disjunctions which are in some sense

near-optimal. In this section, we compare Transformers against those algorithms and also show that they are capable of representing such algorithms.

Classes such as Conjunctions and Disjunctions are known to be efficiently PAC-learnable with $O(n)$ examples. The classical algorithm (see Algorithm 1) for learning Conjunctions works as follows, it starts with a hypothesis $h$ which has all $2n$ literals ($h = z_1 \wedge \bar{z}_1 \wedge z_2 \wedge \bar{z}_2 \wedge \cdots \wedge z_n \wedge \bar{z}_n$). It goes through each example and only checks the positive examples. For each positive example, it finds the literal which are not satisfied and drops those literals from the hypothesis $h$. This is because we know that if a literal is not satisfied and the label of the input is 1 then the target conjunction does not have that literal. For any set of examples, this algorithm returns a conjunction consistent with all examples and PAC-learns the class Conjunctions. A similar algorithm can be derived for Disjunctions as well. See Valiant (1984) for more details.

---

**Algorithm 1** CONJUNCTIONS Classical Algorithm

---

1: **Input:** $n$, $S = ((\mathbf{x}_1, y_1), \ldots, (\mathbf{x}_m, y_m))$
2: **Initialize:** $h = z_1 \wedge \bar{z}_1 \wedge z_2 \wedge \bar{z}_2 \wedge \cdots \wedge z_n \wedge \bar{z}_n$
3: **for** $i = 1, \ldots, m$ **do**
4:     Pick $(\mathbf{x}_i, y_i)$ from $S$
5:     **if** $y_i = 1$ **then**                                     ▷ Ignore negative examples
6:         **for** $j = 1, \ldots, n$ **do**
7:             **if** $x_{i,j} = 0$ **then**
8:                 Drop $z_i$ from $h$         ▷ $j^{\text{th}}$ bit of $i^{\text{th}}$ instance is 0
9:             **else**
10:                Drop $\bar{z}_i$ from $h$       ▷ $j^{\text{th}}$ bit of $i^{\text{th}}$ instance is 1
11:             **end if**
12:         **end for**
13:     **end if**
14: **end for**
15: **Output:** $h$

---

Figure 13 compares the performance of the classical algorithm for Conjunctions (Algorithm 1) and Disjunctions with the performance of Transformers. There are a few things to note here: (a) These classical algorithms are near-optimal and are guaranteed to work for any distributions over the input. They are optimal in the sense that the lower bound on the sample complexity of learning Conjunctions/Disjunctions is $O(n)$ and these PAC-learn these classes with $O(n)$ samples. (b) Transformers can perform better on the first few examples because they have knowledge about the distributions of inputs and functions since they are exposed to it in the meta-learning stage. So in that sense, they are closer to the Bayes-optimal-estimator (Ahuja et al., 2023) than the classical algo in their behaviour. However, unlike the provable PAC-learning algorithms, the algorithms learned by Transformers are not guaranteed to work for arbitrary distributions. (c) Both Transformers and the classical algo reach (near) perfect accuracy after observing almost the same number of $O(n)$ examples.

### E.1 REPRESENTATIONAL COMPLEXITY OF LEARNING CONJUNCTIONS

Interestingly, we can show that Algorithm 1 can represented as a Boolean circuit in $\text{AC}^0$. The class $\text{AC}^0$ contains Boolean circuits with constant depth and a polynomial number of AND ($\wedge$), OR ($\vee$), and NOT ($\neg$) gates.

**Lemma E.1.** *The classical PAC-learning algorithm for learning Conjunctions (Algorithm 1) can be represented as a Boolean circuit in $\text{AC}^0$.*

*Proof.* Essentially what the algorithm is doing is identifying which literal ($z_i$ or $\bar{z}_i$) is definitely not in the target conjunction. The evidence for $z_i$ not being in the target is a positively labelled example $\mathbf{x}$, where $x_i = 0$, i.e. if the $i$-th bit is 0, any conjunction that has $z_i$ as a literal cannot be satisfied by that assignment. Similarly, $\bar{z}_i$ cannot be in the target conjunction if there is a positively labelled example $\mathbf{x}$, with $x_i = 1$. So suppose we ignore the negative examples, and let $S^+ = \{\mathbf{x} \mid (\mathbf{x}, y) \in S, y = 1\}$,

then the truth value of,

$$\bigwedge_{\mathbf{x} \in S^+} x_i,$$

determines whether the literal $z_i$ is in the hypothesis conjunction $h$ output by the algorithm. Likewise, the truth value of,

$$\bigwedge_{\mathbf{x} \in S^+} \bar{x}_i,$$

determines whether the literal $\bar{z}_i$ is in the hypothesis conjunction $h$ output by the algorithm. Thus, using one layer of AND gates, one can determine which literals the algorithm picks to be in the hypothesis conjunction $h$. Let $P_1, \ldots, P_n, N_1, \ldots, N_n$ denote the outputs of these AND gates, i.e. $P_i$ ($N_i$) indicates whether the algorithm puts the literal $z_i$ ($\bar{z}_i$) in its output hypothesis $h$. Then to make a prediction on a new example $\boldsymbol{x}$, the prediction of the algorithm's output hypothesis on this new example is given by the truth value of,

$$\bigwedge_{i=1}^{n} (\bar{P}_i \vee x_i) \wedge \bigwedge_{i=1}^{n} (\bar{N}_i \vee \bar{x}_i).$$

Thus, the entire process of running the algorithm to obtain $h$, and using $h$ to predict the label of a new example $\mathbf{x}$ can be done using an $\text{AC}^0$ circuit of depth at most 3.

□

Recent works (Angluin et al., 2023) have shown that hard-attention Transformers can represent functions in the class $\text{AC}^0$ and since we find that the learning algorithm for Conjunctions is in $\text{AC}^0$ (Lemma E.1), it follows that even hard-attention Transformers can represent the algorithm for learning Conjunctions.

### E.2 How do Transformers learn Conjunctions?

We explore how Transformers could be (in-context) learning Conjunctions in this section. To simplify things, we will focus on Monotone-Conjunctions which contain only positive literals and no negative literals. That is, a function such as $f = z_1 \wedge z_3$ is in the class of Monotone-Conjunctions but $\bar{z}_1 \wedge z_3$ is not since $\bar{z}_1$ is a negative literal.

**Approach.** The key observation behind our approach to probing Transformers is the observation that Monotone Conjunctions can be exactly learned by membership queries alone. In other words, if there is a monotone-conjunction $f$ over $\{0,1\}^n$ which is unknown but we can query the label $f(\mathbf{x})$ for any input $\mathbf{x}$, then we can exactly identify the Conjunction by querying $n$ examples.

The $n$ examples $\mathbf{e}_1, \ldots, \mathbf{e}_n$ will be such that the example $\mathbf{e}_i$ has value 0 at the $i$th coordinate and value 1 everywhere else. For instance, over 4 dimensions, the vector $\mathbf{e}_1 = [0111]$, $\mathbf{e}_2 = [1011]$ and so on. See that for any unknown conjunction $f$ over $\{0,1\}^n$, if we query the function for input $\mathbf{e}_i$ and the label $f(\mathbf{e}_i) = 0$, then it implies that the literal $z_i$ is in the conjunction $f$. Hence, by querying all $n$ such examples $\mathbf{e}_1, \ldots, \mathbf{e}_n$, we can identify any unknown monotone conjunction.

**Interpreting Transformers.** We train the Transformers for the class of monotone conjunctions by sampling prompts and optimizing the model similar to previous experiments (as described in Section 2). We are interested in figuring out what kind of algorithm it applies to learn conjunctions in-context. During evaluation, the model $M$ sequentially receives prompts of the form $P_k = (\mathbf{x}_1, y_1, \ldots, \mathbf{x}_{k-1}, y_{k-1}, \mathbf{x}_k)$ for $1 \le k \le m$. After receiving each prefix $P_k$, we evaluate the model $M$ on the $n$ examples $\mathbf{e}_1, \ldots, \mathbf{e}_n$ to identify the function used by the Transformer model $M$. More specifically, let $P_k^{(i)} = (\mathbf{x}_1, y_1, \ldots, \mathbf{x}_k, y_k, \mathbf{e}_i)$, then for each $k \in [0, \ldots, m]$, we evaluate the model on $P_k^{(1)}, \ldots, P_k^{(n)}$.

See that if the Transformer model $M$ makes its prediction based on some unknown Conjunction, then we can recover it during each step of in-context learning. However, it is not necessary that the model $M$ predicts according to a conjunction and may be using some other form of function in which case this approach is not guaranteed to work. Note that, since Transformers reach perfect accuracy after observing a certain number of examples (Figure 6 top-left) and also achieve that on

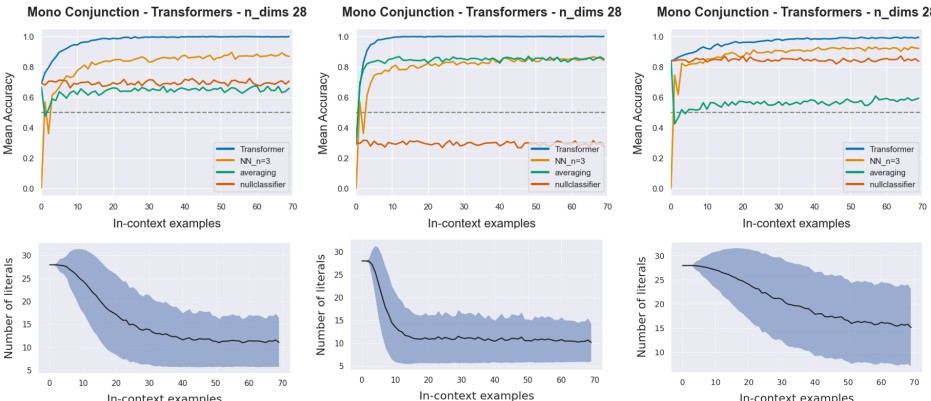

Figure 16: Extracting the target (Mono) Conjunction from Transformer during in-context learning. The upper row depicts Transformer's performance across in-context examples and the lower row shows the number of literals in the conjunction extracted from the Transformer at that point. The values are averaged across 1k prompts. The three columns are for three different distributions over inputs (at test time). First column: $\approx 30\%$ of inputs in the prompt are positive, second: $\approx 70\%$ of inputs are positive, third: $\approx 15\%$ of inputs are positive.

out-of-distribution examples, it is reasonable to expect it to predict according to some underlying conjunction as it sees more examples. Nonetheless, we apply the approach described above to extract the target Conjunction at each step of the in-context learning process.

**Learning Mono Conjunctions.** For Monotone-Conjunctions, the PAC-learning algorithm is a simpler version of Alg. 1. It starts with a conjunction with all literal $h = z_1 \wedge z_2 \cdots \wedge z_n$ and while going through $m$ examples, it only looks at the positive examples. For any positively labelled example $x_i$, if any coordinate $x_{i,j} = 0$, then it drops the literal $z_j$ from the hypothesis $h$ and outputs $h$ after going through all $m$ examples.

**Results.** We find that Transformers behave in a manner very similar to the algorithm described above but not in an exact way. We evaluate the model on 1k prompts containing 70 examples each and extract the literals at each step beginning from no examples to all 70 examples. We observe that in the beginning, the target function has all literals similar to the classical algorithm. Moreover, as the model observes more examples, the number of literals decreases monotonically and finally converges to the correct target conjunction which was used to label the examples in the prompt. However, unlike the classical algorithm the Transformer model does not drop literals right after seeing a positive example, we find that it drops a larger number of literals together after observing a series of positive examples.

Figure 16 depicts the number of literals in the extracted conjunction across in-context examples for three different input distributions. The model was trained according to the first distribution where approximately $30\%$ of the examples in the prompt are positively labelled and then tested on three different distributions. Similar to the classical algorithm, the model seems to start with all literals and then gradually drops literals as it sees more examples. Upon manual inspection, we found that the model does not always drop literals right when it sees a positive example. At the same time, as can be seen from Figure 16, when the prompt contains more positive examples (second column), the model drops literals more quickly and reaches perfect accuracy with fewer examples overall. On the other hand, with fewer positive examples, the Transformer seemingly drops literals later (third column in Figure 16) and needs more examples to reach near-perfect accuracy. This behaviour is similar to the classical algorithm. Additionally, we compute the number of times the Transformer model drops an incorrect literal (a literal which is present in the target conjunction) at any point in time during the in-context learning phase. We find that across all three input distributions, it drops a literal incorrectly in less than $0.1\%$ out of 1k prompts containing 70 examples each. In over $95\%$ of the prompts, it converges to the correct target conjunction. Our results suggest that Transformers implement an algorithm akin to the classical one, initially starting with all literals and then progressively dropping

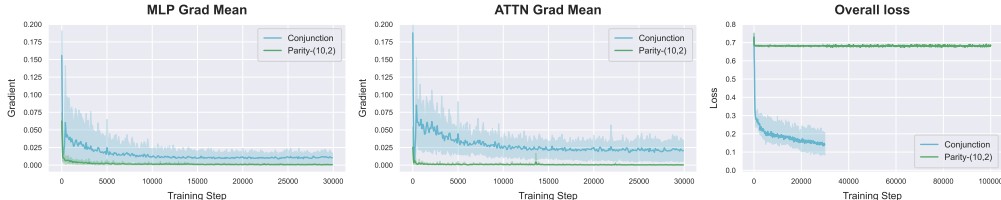

Figure 17: *Left:* Figure depicting the mean of the gradients of parameters such as MLP and attention parameters across different iterations of training on Parity-(10, 2) and Conjunctions. *Right:* The loss of Transformers on Conjunctions vs Parity-(10, 2) across different iterations of training.

them as they encounter more and more positive examples. However, deviations from the classical algorithms exist which remain to be understood.

## F   CURIOUS CASE OF LEARNING PARITIES

An interesting and somewhat surprising phenomenon is the failure of Transformers and other architectures on the class of Parities and sparse Parities. In contrast to other function classes where models are able to learn some learning algorithm, for Parities, they do not seem to be able to find any kind of learning algorithm that performs beyond chance-level accuracy.

We examine the gradients and the model parameters during the training process. In Figure 17, we show the mean of gradients for all parameters of attention and feedforward networks in the Transformer. Observe that the gradients are very close to $0$ after a few iterations of training. Note that this is in contrast to what we observe while training Transformers for tasks such as Conjunctions where gradients are not so close to $0$. However, the exact reason why Transformers fail to learn anything is unclear and is left as an open problem.

The problem of learning Parities and Sparse Parities have well-known hardness results Kearns (1998) in the Statistical Query framework indicating that Parities require $2^{\Omega(n)}$ queries and Sparse Parities require $n^{\Omega(k)}$ queries (where $k$ is the number of relevant bits). Gradient-based methods are considered to be only as powerful as SQ-learning algorithms in finite-precision settings and hence any model such as Transformers applying a gradient-based algorithm to learn in-context would require an exponential number of steps to predict accurately. For sparse parities PARITY-$(10, 2)$, we provide more than $n^k$ examples in the prompt and hence a model applying gradient-based learning such as FFNs can solve it (see Fig. 18 Left). For instance, feedforward networks trained with gradient descent on the same number of examples as provided to Transformers for ICL are able to achieve near-perfect accuracy. However, Transformers and other architectures do not seem to make any progress during their training process. Also, note that the hardness of learning Parities primarily holds for uniform distribution over inputs and does not apply to input sets with teaching sequences which are no longer uniform.

Parities are PAC-learnable using the Gaussian elimination algorithm (Fischer & Simon, 1992) with $O(n)$ examples and if any of the architectures could learn to implement the algorithm, then they would have been able to solve it with $O(n)$ examples.

**Transformers vs LSTMs.** An interesting difference we observed is that for the PARITY-$(10, 2)$ problem, LSTMs achieve $100\%$ accuracy if the test examples are labelled by the same set of functions as the training examples. This is not true for Transformers. For the PARITY-$(10, 2)$ problem, there are only $45$ different functions. For the results in Table 1, we ensure that the training examples are labelled by 23 of those functions and the test examples are labelled by the remaining and unseen functions. However, if we uniformly sample a function out of the $45$ functions during both training and evaluation, then we observe that LSTMs achieve perfect accuracy during evaluation. This is interesting since the sequence of input points $x_1, \ldots, x_k$ are new during evaluation and hence is not a clear case of memorization. On the other hand, since they do not generalize to unseen functions, it is difficult to claim that LSTMs can learn sparse parities in-context.

**Curriculum learning.** For Sparse Parities, we also tried curriculum learning similar to Garg et al. (2022) but did not observe any improvement. In particular, we trained Transformers for the problem

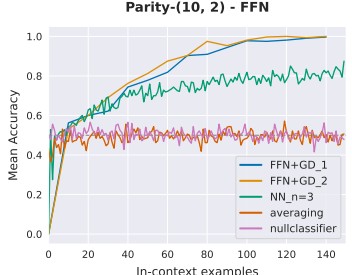 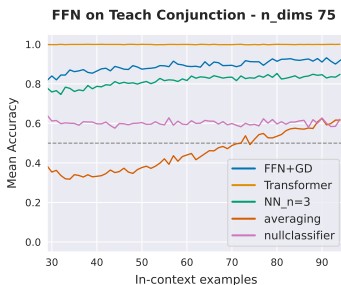

Figure 18: Left: Performance of Feedforward networks (FFN) trained with gradient descent on Parity-(10, 2) with the same number of examples as provided to models in ICL setting. *Right:* Performance of FFN on Teach Conjunction task.

of PARITY-$(14, 2)$. In the curriculum learning setup, the input dimension is initially smaller, i.e., $9$ and it increases incrementally by 1 after every 50k steps. Hence, effectively for the first 50k steps, the model is trained for PARITY-$(9, 2)$, then PARITY-$(10, 2)$ and so on eventually ending at $14$ (trained 250k steps after reaching 14). The model is trained for 500k steps in total. The number of examples in the prompt $m$ is 120 in the beginning and increases by 20 every 50k steps and eventually stays at 200. We start at input dimension 9 because below that the total number of Boolean inputs is very small and the prompt will cover over half of all possible inputs. We do not find any improvement with the curriculum learning setup as well. The loss curve is identical to the one in Figure 17 (right) and is hence omitted.

**Performance of Nearest Neighbours.** In the case of sparse parities, nearest neighbour approaches can do quite well, particularly for relatively small values of $k$ and $n$. For instance, suppose there are only two relevant bits, say $k = 2$, then for any fixed point $\mathbf{x}$, conditioned on the relevant bits matching, the expected distance to a random point is $(n - 2)/2$. On the other hand, if the relevant bits don't exactly match, then the expected distance to a random point is at least $(n - 2)/2 + 1$. Since the probability that the two relevant bits match is $1/4$, we expect with a relatively high probability that the nearest neighbour would match the relevant bits. As a result the nearest neighbour classifier will have a reasonably high accuracy. We observe that the performance of LLMs on this task is better than random, and comparable to nearest neighbour.

## G TEACHING SEQUENCE EXPERIMENTS: ADDITIONAL DETAILS

**Teaching Sequences.** The concept of teaching sequences and teaching dimensions was introduced in Goldman & Kearns (1995). For any class of functions $\mathcal{F}$, the teaching sequence of a function $f$ in $\mathcal{F}$ is a sequence of labeled instances $(\mathbf{x}_1, \mathbf{y}_1, \ldots, \mathbf{x}_m, \mathbf{y}_m)$ such that it unambiguously identifies $f$ in $\mathcal{F}$ – only the function $f$ is consistent with the sequence of labeled instances. Let $\mathrm{T}(f)$ be the set of all teaching sequences for the function $f$. The teaching dimension (Goldman & Kearns, 1995) of a function class is defined as, $\mathrm{TD}(\mathcal{F}) = \max_{f \in \mathcal{F}}(\min_{\tau \in \mathrm{T}(f)} |\tau|)$.

**Conjunctions and Disjunctions.** The teaching sequence for Conjunctions can be created as follows. Suppose $f$ is a conjunction of $k$ literals. We will create $k + 2$ points. For the first two points, the variables corresponding to all literals in the Conjunction are set to True, that is, they are 1 for positive literals in the Conjunction and 0 for negative literals. In the first point, all variables that do not affect the output of the function are set to 0 and in the second point, all of them are set to 1. The next $k$ points will correspond to each of the literal in $f$. For each literal $z_i$, we create a point $x \in \{0, 1\}^n$ where the variable of $x$ at $i$ is 0 if $z_i$ is in the Conjunction and is 1 is $\bar{z}_i$ is in the Conjunction. The variable corresponding to every other literal in $f$ is set to 1 if it is a positive literal and is 0 if it is a negative literal. The variables which do not affect the output of the function are all set to 0. Essentially, these examples show that even if the variables corresponding to every other literal in the function are set to true, the function value will still be 0 because one of the literal values is set to false. The teaching sequences for Disjunctions are created in a similar manner.

See that since for every Conjunction or Disjunction, the length of the teaching sequence is $k + 2$, the teaching dimension is $n + 2$ and for each function with $k$ literals, the exact function can be identified

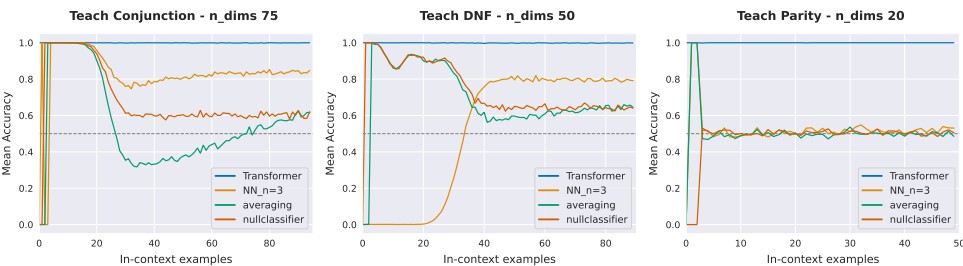

Figure 19: Performance of Transformers on various tasks with Teaching Sequences. The plots depict the performance of models on all examples. Note that, most examples in the teaching sequence have the same label and hence even the Nullclassifier has near-perfect accuracy. The plot values up to the expected length of the teaching sequences are omitted to avoid confusion. Refer to Section G.1 for more details.

with $k + 2$ points in the prompt example or a training set. Note that the sample complexity of $O(n)$ for Conjunctions or Disjunctions is with respect to PAC-learning in the sense that with $O(n)$ points, one could guarantee that the error will be within some $\epsilon$ for arbitrary distributions. However, the exact number of samples will depend on the value of $\epsilon$ and could be much larger than $k$ or $n$ when the error $\epsilon$ is required to be very small.

**DNFs and CNFs.** For simplicity let's consider Monotone 3-DNFs which only have positive literals. For each clause or Conjunction in the DNF, we create $k + 1$ points. The first point is such that variables corresponding to every literal in the clause are set to $1$ and all other variables are $0$. The other $k$ points are created as follows. For every literal in the clause, there is one point where the variable corresponding to the literal is $0$ and the variables corresponding to other literals in the clause are set to $1$. The variables which are not in the clause are set to $0$. For instance if the input is over $\{0, 1\}^4$ and the clause is $x_1 \wedge x_3$, then the three points corresponding to the clause will be $[1010], [0010]$, and $[1000]$. The teaching sequence for the 3-DNF is the union of the teaching sequence of its three clauses and hence will have size $l + 3$ where $l$ is the total number of literals in all clauses. The teaching sequence for 3-CNF can be created in a similar manner.

**Sparse Parities.** For sparse parities with $k$ relevant bits, the teaching sequence has $k$ points where in each point the variable corresponding to one of the relevant bits is set to $1$ and every other bit in the input is set to $0$. Note that, this is a teaching sequence of the class PARITY-$(n, k)$ which have exactly $k$ relevant bits. This would not work as a teaching sequence for the entire set of Parity functions PARITY-$n$.

For further details on teaching sequences, see Goldman & Kearns (1995).

### G.1    ADDITIONAL DISCUSSION

**Performance of FFN.** Figure 18 (right) depicts the performance of feedforward networks trained with gradient descent on examples with the teaching sequence. FFNs require a larger number of additional points apart from the teaching sequence to predict with high accuracy. This is not quite surprising as FFNs+SGD are quite general-purpose learning algorithms. At the same time, it is difficult to integrate such problem-specific knowledge into them so that they perform more optimally.

**Additional Details of Plots.** Figure 19 depicts the performance of Transformers and baselines on all examples including the teaching sequences. Most examples in the teaching sequence have the same label and hence even the Nullclassifier obtains near-perfect accuracy for some cases. If the structure of the teaching sequence is known and it is known that the first $t$ examples will be the teaching sequence, then it is also possible to predict the labels of the teaching sequence correctly. For sparse parity, all examples in the teaching sequence have the label $1$. For Conjunctions and DNFs, only a couple of examples have the label $1$ and all other examples have the label $0$. Such examples are visibly distinct and hence it is straightforward to predict their label if it is known that they are part of the teaching sequence. The plot values up to teaching sequences are omitted in the main paper to avoid confusion since they do not reflect the effectiveness of the model or the baselines on the tasks.

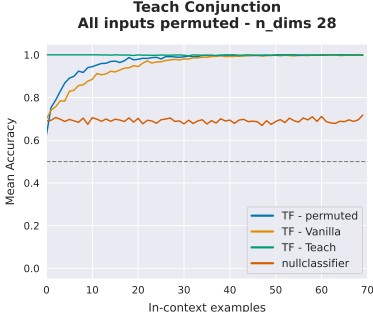 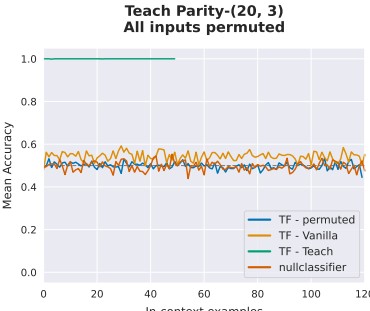

Figure 20: Performance of Transformers on prompts with a permutation of teaching sequence examples and random examples. On Sparse Parities, the behaviour is the same as the vanilla case whereas for Conjunctions, the performance improves for prompts with 70 examples. The setting becomes closer to the vanilla case as the prompt length goes higher. See Section G.1 for more details.

**Performance on OOD functions.** We explored the performance of Transformers on out-of-distribution prompts for Conjunctions where the teaching sequences are typically shorter or longer compared to the distribution during training. The distribution over Conjunctions during training is the same as the vanilla setting described in Section C.2. The probability of any literal (or its negation) being in the target conjunction is $30\%$ and hence for inputs over $n = 75$ dimension, the expected length of the teaching sequence is $27$. We evaluated the trained Transformer on prompts from two different distributions. In the first distribution, the probability of literals being in the Conjunction is $60\%$ and hence the expected length of teaching sequences is $47$. Similarly, in the other distribution, the probability of literals being in the Conjunction is $15\%$ and hence the expected length of teaching sequences is $\approx 13$. In both cases, the behaviour of the trained Transformer is identical to the in-distribution setting. The evaluation curve is identical to Figure 19 and is hence omitted.

**Permuted Examples.** We have considered two types of prompts: (a) prompts with random examples sampled from some distribution and (b) prompts which begin with a teaching sequence followed by random examples. We explore the performance of Transformers on another type of prompt which includes both examples from teaching sequences and random samples. In this third type (c) called 'permuted', for a prompt with $m$ total examples, we create the teaching sequence with $t$ examples and sample $m - t$ examples from a distribution (given in Section C.2) and permute the entire set of inputs to create the prompt. In other words, the examples in the teaching sequence are present in the prompt but not necessarily in the beginning anymore.

Figure 20 depicts the performance of Transformers on the three types of prompts respectively (a) prompts with random examples (marked vanilla), (b) prompts with teaching sequence in the beginning (marked teach) and (c) prompts with a permuted combination of teaching sequence and random examples (marked permuted). Without the teaching sequence in the beginning one would naturally expect the setting to be closer to the vanilla case – which happens in the case of Sparse Parities (Figure 20). For Conjunctions, the behaviour is relatively more interesting. Unlike Sparse Parities, some examples of the teaching sequence of Conjunctions are more informative than others. With 70 examples, Transformers in the vanilla setting reach near-perfect accuracy with about 30 examples. Since the likelihood of those couple of examples in the teaching sequence being in the first 30 examples is reasonably high, Transformers with such permuted prompt examples seem to converge to high accuracy faster than the vanilla case. As the size of the prompt is increased from 70, the behaviour becomes closer to the vanilla case since the probability of the informative examples being in the first 30 decreases.

# H  Additional Details of Experiments with LLMs

## H.1  Additional Details of Frozen GPT Experiment

In Section 5, we discuss the setup and results for experiments with pretrained models. We provide additional details of the experiments here.

You are given some examples of inputs and their corresponding labels. You need to learn the underlying boolean function represented by these input-label examples. Predict the label (either 0 or 1) for the final input.

Input: 0 0 0 1 0
Label: 0
Input: 1 0 0 0 1
Label: 0
Input: 0 0 0 0 1
Label: 0

(. . . more exemplars . . . )

Input: 1 1 1 0 1
Label: 1
Input: 1 1 1 0 0
Label: 0
Input: 0 1 1 0 0
Label:

Figure 21: An example of a prompt provided to the LLM in the direct evaluation experiments. Here, the model has to learn a function from the Majority class in 5 dimensions and predict the label for the last input. The prompt consists of a natural language instruction and $k$ example points of the function.

In Section 3 and 4 as well as several prior works (Garg et al., 2022; Ahuja et al., 2023; Bai et al., 2023), experiments were conducted in the meta-learning-like setup in order to understand the in-context learning abilities of Transformers. With the experiments with frozen pretrained models, we aim to explore if pretrained models learn any mechanism that is useful for solving tasks in the meta-learning-like stylised setup.

The experiments follow the same framework as described in Section 2 with the exception the weights of the Transformers are not modified or updated during the training process apart from the input embedding layer and the final output layer. We use the GPT-2 XL (1.5B parameters) model from Hugging Face (Wolf et al., 2020).

In the stylised setup, since each Boolean input $x \in \{0, 1\}^n$ is provided as a single token, the original embeddings and tokenizer of the pretrained model cannot be used for prediction. Hence, we replace the original embedding layer with a 1-layer ReLU FFN: $\{0, 1\}^n \rightarrow \mathbb{R}^{\text{width}}$ which produces an embedding for each $x \in \{0, 1\}^n$. Similarly, we add an output FFN which predicts the final output given the vector produced by the pretrained model.

Similar to other experiments we train the model on Conjunctions and the nearest neighbour task by sampling functions and input points to create training examples. The training process is used to update the embedding and output layer and the weights of the Transformers are fixed.

## H.2    DETECTING NEAREST NEIGHBOUR HEADS IN GPT-2

In Section 5.1, we discussed that we were able to discover attention heads in the GPT-2 model that very closely implemented the nearest neighbours algorithm. In this section, we provide more details about that experiment.

Our methodology for detecting attention heads that directly implement the nearest neighbours algorithm is similar to the one used by Olsson et al. (2022) for finding induction heads. While training the model on the Nearest Neighbours task, for each attention head in the model, we compute a 'nearest-neighbours score (nn-score)'. After every 500 training steps, we provide the model with a batch of 100 examples, each example $e_i$ consisting of 80 points, i.e., $e_i = (\mathbf{x}_1, \mathbf{y}_1, \ldots, \mathbf{x}_{80}, \mathbf{y}_{80})$. The nn-score is the attention paid to $\mathbf{y}_j$ for predicting $\mathbf{y}_k$ ($0 < j < k$) where, $j = \arg\max_{i=0}^{k-1} \left( \frac{\mathbf{x}_i^T \mathbf{x}_k}{\|\mathbf{x}_i\|\|\mathbf{x}_k\|} \right)$ is the index of the nearest neighbour of $\mathbf{x}_k$, averaged for the last 40 points (i.e., $k > 40$) over all 100

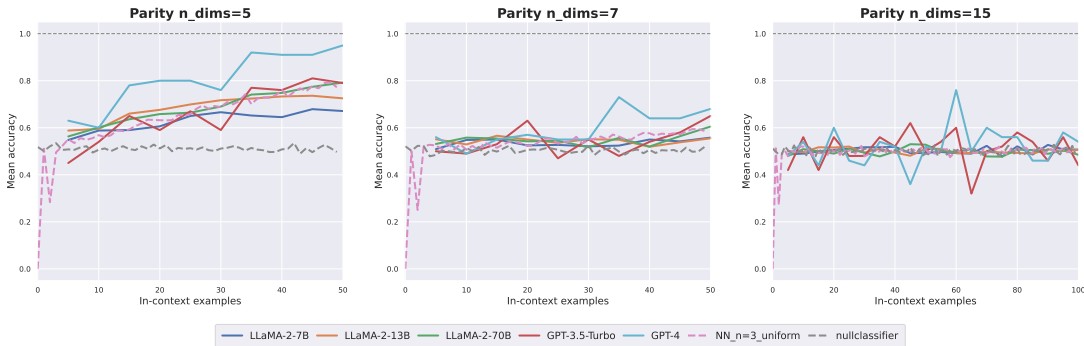

Figure 22: Results with the direct evaluation of LLMs at inference time across varying dimensions for the PARITY-$n$ task.

examples. If the nn-score for a particular head is greater than 0.5, we label that head as a 'nearest neighbour head'. In our experiments with frozen GPT on the Nearest Neighbours task, we were able to consistently find 8-10 nearest neighbour heads across multiple runs.

### H.3    ADDITIONAL DETAILS AND RESULTS OF DIRECT EVALUATION EXPERIMENT

In Section 5, we described the setting of direct evaluation of LLMs. We provide additional details regarding the experimental setup below.

The main prompt (see Figure 21 for an example) comprises of an instruction in natural language followed by a series of $k$ example points $(\mathbf{x}_i, \mathbf{y}_i)$, where $\mathbf{x}_i \in \{0,1\}^d$ is provided as a sequence of tokens $x_1, \ldots, x_d, x_j \in \{0,1\}$ and $\mathbf{y}_i \in \{0,1\}$ is also provided as a single token. The model's goal is to predict the correct label $\mathbf{y}_{k+1} \in \{0,1\}$ for the query point $\mathbf{x}_{k+1}$. For a particular function, we sample a sequence of $n$ points and prompt the model with $k < n$ points as in-context exemplars of the function. We call the model multiple times, increasing $k$ by a value of 5 every time. The prompt in each call consists of the first $k$ points $(5 < k < n)$ from the $n$-point sequence, and asks to predict the $(k+1)^{\text{th}}$ point. We repeat this experiment with 100 different functions[3] from the same function class.

We experiment with three classes of functions: Conjunction, Majority, and Parity. We consider dimension $d \in \{5, 7, 15\}$ and set the number of points $n = 50$ when $d < 10$ and $n = 100$ when $d \geq 10$.

**Additional Results.** The results of evaluating LLMs directly on the Parity task are provided in Figure 22. We also experimented with LLaMA-2 models of smaller scale as well as GPT-2 as shown in Figure 23. Our results seem to indicate that model scale seems to play some role in the ability of LLMs to implement learning algorithms. This can be seen from the performance of the pre-trained GPT-2 model, which fails on both tasks even for 5 dimensions, and the (gradual) increase in performance as we increase the size of the LLaMA models. Also, it is quite surprising to see that even smaller LLaMA models are also able to achieve nontrivial levels of performance for both tasks.

## I    IMPLEMENTATION DETAILS

All of our implementations for experiments are based on PyTorch (Paszke et al., 2019). For our experiments with Transformers trained from scratch, we use the Huggingface Transformers library (Wolf et al., 2020) and our own custom implementation. For recurrent models such as LSTMs, we use PyTorch's inbuilt implementation. For Hyena (Poli et al., 2023) and DSS (Gupta et al., 2022), we adapt the official implementation by the authors for our experimental setting. For RetNet (Sun et al., 2023), we use our own implementation.

---

[3]It is prohibitively expensive to experiment with a very high number of functions for the OpenAI models, so we limit the number of functions to 100. We average over 2000 different functions for the baselines and 1000 different functions for the LLaMA-2 models for more robust results.

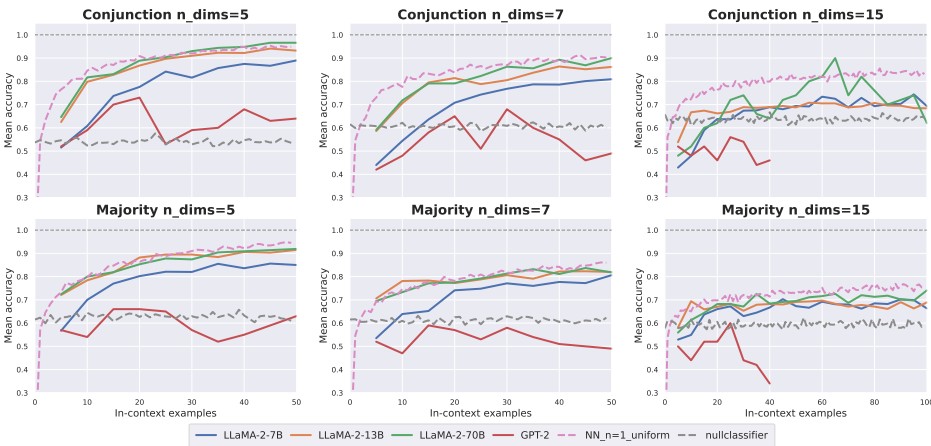

Figure 23: Results with direct evaluation for LLaMA models of different scales and GPT-2. The *top* row shows the performance across varying dimensions for the Conjunction task while the *bottom* row shows the performance for the Majority task.

| Hyperparameter | Transformer and others | LSTM |
|:---:|:---:|:---:|
| D_model/Hidden Size | [256, 512] | [64, 512] |
| Heads | [4, 8 ] | - |
| Order | [2, 3] | - |
| Number of Layers | [1, 16] | [1, 8] |
| Learning Rate | [1e-2, 1e-5] | [1e-2, 1e-5] |
| Position Encoding Scheme | [Learnable, Absolute] | - |

Table 3: Different hyperparameters and the range of values considered for each of them. The hyperparameter *Heads* is only relevant for Transformers and RetNets. The hyperparameter *Order* is only relevant for Hyena. See Section I for more details.

All our experiments were conducted using 16 NVIDIA Tesla V100 GPUs each with 16GB memory. For each dataset, we extensively tune across several hyperparameters and report the results based on the best-performing models. Table 3 lists the hyperparameters used for tuning the models for Boolean function experiments in Section 3. We use a grid search procedure to tune the hyperparameters. For all our results, we used Adam Optimizer and tuned the learning rates. For all architectures apart from LSTMs, we tuned across depths $\in \{1, 2, 6, 12, 16\}$ and for LSTMs we used depths $\{1, 2, 3, 8\}$ primarily because LSTMs with large depths are harder to train. For LSTMs we tuned the width or hidden_size across $\{64, 256, 378, 512\}$ and for other architectures we use $\{256, 512\}$. For Hyena, we tune the order hyperparameter across $\{2, 3\}$ and for Transformers and Retnets we tune the heads across $\{4, 8\}$. For all non-recurrent models, we try both learnable and absolute positional embedding schemes. For all models, we tune the learning rate across $\{0.01, 0.005, 0.001, 0.0005, 0.0001, 0.00005, 0.000001\}$. We train the models for different numbers of steps/iterations $\in [30k, 500k]$ depending on the task where each iteration is with a fresh batch of in-context sequences. Note that in all these tasks apart from the experiments with finite samples in Section D.3, the training loss never reduces to 0 since each batch of examples is labeled by a different function (with high probability). We train for a large number of steps (200k) if the loss doesn't reduce or train till the loss has reduced and converged for at least 20k steps. For instance, for tasks like Conjunctions models converge to low loss in 20k steps so we train the models to up to 40k steps. For other tasks such as Parities, we train for 500k steps. For the additional linear regression task (Figure 8, we follow the same setup and hyperparameters as described in Garg et al. (2022).

Based on the experiments, we find that Transformers and Hyena are relatively much more robust across various hyperparameters than other architectures. Among most tasks and architecture, we find that the learning rate has the most influence on the performance and the depth has a nontrivial influence. For other hyperparameters such as heads and widths, we do not see any nontrivial difference in performance.

**Direct Evaluation Experiments.** For experiments with open-source models such as LLaMA-2, we use Huggingface Text-Generation-Inference[4]. Experiments using GPT-3.5-Turbo and GPT-4 were performed using the OpenAI API[5]. Experiments with locally deployed large models like LLaMA-2 were conducted using 2 A100 GPUs each with 80GB memory. For all experiments, we decode greedily and generate a maximum of two tokens (whitespace and the label[6]).

## J   ADDITIONAL RELATED WORK

Our results on the adaptive selection of algorithms add to some observations in recent works. Ahuja et al. (2023); Bai et al. (2023) explored (among other things) the efficacy of Transformers in learning mixtures of tasks in-context and showed that they could adaptively select appropriate algorithms depending on the sequence of inputs and labels. In our case, the emphasis is on choosing between different algorithms for the *same class of functions* (e.g. Conjunctions). Hence, the target labelling function is still a Conjunction where the vanilla algorithm leads to near-perfect accuracy as well but models such as Transformers are able to choose the more sample-efficient algorithm which leads to near-perfect accuracy with fewer input examples. The fact that LSTMs can exhibit in-context learning phenomenon was observed in Xie et al. (2021) as well. Our work attempts to conduct a more systematic comparison of multiple architectures on a test bed with a variety of tasks. Evidence from our experiments suggests that while attention may not be necessary and various attention-free models can solve multiple in-context learning tasks, they are not sufficient at the moment to match Transformers' performance on all tasks considered in our work.

In recent years, there have been many empirical works (Liu et al., 2021; Min et al., 2021; Lu et al., 2022; Razeghi et al., 2022; Min et al., 2022a; Olsson et al., 2022; Wei et al., 2023) that have analysed the capabilities and limitations of in-context learning (ICL) in large language models (LLMs). The ability of LLMs to solve simple prediction problems has been informally explored in Lovre (2022). Liu et al. (2021) explored various strategies for selecting examples that improve in-context learning performance. Lu et al. (2022) demonstrated that the in-context learning ability of models is also sensitive to the order in which the samples are provided in the prompt. Razeghi et al. (2022) worked with numerical reasoning tasks and showed that in-context learning performance is stronger for instances whose terms are more prevalent in the training data. Min et al. (2022a) questioned whether models really learned new tasks in-context by showing strong ICL performance even when the prompt labels are chosen randomly. (Olsson et al., 2022) and (Elhage et al., 2021) frame the in-context learning ability slightly differently by referring to any model that gets better (i.e., shows a decrease in loss) with more context (i.e., increasing token indices) as an in-context learner. They provide arguments for the existence of special circuits, which predict the next token by observing similar patterns previously seen in context, inside the Transformer model, that are responsible for in-context learning. More recently, Wei et al. (2023) showed that the ability of models to override semantic priors and effectively learn the task presented in-context emerges with scale.

---

[4]https://github.com/huggingface/text-generation-inference

[5]https://platform.openai.com/

[6]If the model does not predict a label in $\{0, 1\}$, we consider the prediction incorrect.

