# OpenReview forum: "Understanding In-Context Learning in Transformers and LLMs by Learning to Learn Discrete Functions"
_ICLR.cc/2024/Conference — ICLR 2024 oral_

### Official Review · Reviewer_9RLW · 2023-10-18

**Soundness:** 3 good
**Presentation:** 2 fair
**Contribution:** 3 good
**Rating:** 8
**Confidence:** 3

**Summary:**

This paper investigated the in-context learnability of Transformers on discrete Boolean functions. They showed that when trained in an in-context way, the Transformers can in-context learn some Boolean functions like conjunctions, disjunctions, DNFs and CNFs, while they can struggle on more difficult tasks such as parities. They showed that when presented the teaching sequences, the Transformers show better in-context learnability and they can learn tasks like parities in this case.

They also showed that besides TFs, many other architectures show in-context learnability competitive to TFs on learning Boolean functions. They hypothesized that TFs learn two distinct algorithms on some tasks, which I feel not that convincing (I will explain it later). They also did some experiments on LLM and showed that the GPT2 architecture with fixed parameter, and many other LLMs can in-context learn Boolean functions and are competitive to Nearest Neighbour.

In general, this paper showed in experiments that the trained TFs can in-context learn some Boolean functions. The experiments with teaching sequence is particularly interesting. It's possible for me to update the score.

Thanks for the response from the authors. I raised my score to eight.

**Strengths:**

1. The  experiments are sufficient and interesting, showing the in-context learnability of trained TFs on discrete Boolean functions. Previous papers mostly focused on the IC learnability of linear functions, sparse linear functions or NNs. The consideration of Boolean functions is particularly creative.

2. The experiments about teaching sequence is particularly interesting, and also for the experiments on LLMs. The fact that LLMs can implement the kNN algorithm in-context is amazing, and I also like the narratives of the algorithm learning perspective of in-context learning.

3. The writing is great and easy to follow.

**Weaknesses:**

1. The author claimed that TFs can learn two distinct algorithms on tasks such as conjunctions, which is not that convincing to me.

Let's take figure 3 as an example and let's call the four sub-figure a to d from the left to the right. The way that authors compared these four figures and draw conclusions is that: they compare a and b, and the compare c with d, then they drew conclusion that the trained TFs can learn two distinct algorithms. This is not a good way for comparison. This is because one can easily give a counter-example to show that a single algorithm can simultaneously achieve a and b. For example, suppose the algorithm that TFs learn is: randomly pick one decision rule (or Boolean function) which is in the conjunction function class and agrees with all examples in the test context (the x_i, y_i pairs at test time). Then, when tested on teach conjunction (fig a), since there is only one conjunction function matching all test context, it will achieve a perfect accuracy. When tested with random sequence of examples from a conjunction task, there can be many functions in the conjunction function class that satisfying all test examples, so it can achieve a imperfect performance. Similar logic applies on the comparison between c and d.

At a high level, if you want to show the trained TFs can learn two distinct algorithms, it is not a good idea to show that they achieve different performance on different test examples but trained on same data distribution, It is better to show that they achieve different performance when trained on different data distribution, but tested on the same examples. In this case, you should compare figure b and c to draw your conclusion, instead of comparing a and b. The reason behind this is that the in-context learning is specific to the pre-training distribution, and it s very natural that they learn different algorithms on different pre-training distributions.

**Questions:**

Below are my suggestions, which I think can help make your paper stronger.

1. Your definition and narratives of in-context learning and the algorithm learning perspective (the last two paragraphs in 'in-context learning' paragraph in section 2) seems to follow closely with the formulation in [1] and the formal definition in [2]. Does your definition of in-context learning similar to their definitions? Or do you have any differences? IMO maybe you can discuss more about your definition of ICL and how it relates to algorithm learning process.

2. I am not sure how you sampled the random conjunction function for each task. I think it is more important to say how you sample the random functions than how you sample the random inputs. Do you sample the random conjunction functions by sampling k variables for a fixed k and then form the conjunction function as the intersection of these k variables (f = 1 iff all these k variables are 1)?

3. Is it possible for you to show what algorithm the TFs actually implement on some simple tasks like conjunction or disjunction? For this, you can either look into the attention weight of attention matrix (maybe train a single-layer TF?) or maybe you can try to provide some constructions that can provably learn these tasks? These can be hard even an open problem, which is definitely not a requirement.

4. For some tasks in your paper, there should be existing algorithms to learn it, such as conjunction or disjunction. I am also wondering how does trained TFs compare to them? Also, some existing paper shows that the TFs can approximate or possibly learn (in-context) the Bayesian optimal estimator over some function class [1,2,3]. I suppose under the task distribution in your paper, it is very likely that the Bayesian optimal estimator is analytically intractible, but I think it should be numerically computable under some simple tasks. So I am wondering how does trained TFs compare to the Bayesian optimal estimators (or some strong, computationally efficient baseline on specific task, if any).

[1]. Transformers as Algorithms: Generalization and Stability in In-context Learning

[2]. Trained Transformers Learn Linear Models In-Context

[3]. Pretraining task diversity and the emergence of non-Bayesian in-context learning for regression.

---

> ### Author Response · Authors · 2023-11-14
> **Response to Weakness and Questions**
>
> Thank you for your thoughtful comments and time.
>
>
>
> **Response to weakness**
>
> “Reviewer: *The author claimed that TFs can learn two distinct algorithms on tasks such as conjunctions, which is not that convincing to me.
> Let's take figure 3 as an example and let's call the four sub-figure a to d from the left to the right. The way that authors compared these four figures and draw conclusions is that: ….. , so it can achieve a imperfect performance. Similar logic applies on the comparison between c and d.*”
>
> We think there could be a misunderstanding here. We are not completely sure we understand your explanation here. However, based on our interpretation, we think our argument in the paper is quite similar to what you suggest in the next paragraph.
>
> “Reviewer: *At a high level, if you want to show the trained TFs can learn two distinct algorithms, it is not a good idea to show that they achieve different performance on different test examples but trained on same data distribution, It is better to show that they achieve different performance when trained on different data distribution, but tested on the same examples. …*”
>
> We are not comparing Transformers trained on the same data distribution. We are comparing two Transformers trained on two different pretraining distributions (one with random examples and one with teaching sequences). Leaving aside Transformers, it is straightforward to see that there can be two types of algorithms: algo (a) is a more general purpose like FFN trained with gradient descent and algo (b) is a more specific algorithm which uses teaching sequences to find the correct conjunction. Now, the general purpose algo (a) will perform well with and without teaching sequences but it need not be optimal on examples with teaching sequences. On the other hand, the algo (b) will be optimal on examples with teaching sequences but can completely fail when such teaching sequences are not provided.
>
> We claim that Transformers learn two distinct algorithms in this sense. This part is unrelated to Figures 3 c and d. If you look at Figure 3 a and Table 1 (or Figure 6) then these correspond to results with Transformers trained for Conjunctions in the vanilla setting (without teaching sequences) which work well with and without teaching sequences (analogous to aglo (a) described above). Figure 2 left and Figure 3 c (center-left) correspond to results with Transformers trained with teaching sequences (different pretraining distribution). Figure 2 left shows that they can perform optimally when teaching sequences are provided but fail otherwise (Figure 3 c) (analogous to algo (b)).  Figure 3 a and b are results with two different Transformers trained on two different input distributions (mentioned in the title above the figure). Figures 3 c and d serve a different purpose which depicts the result of a single Transformer trained on a mixture of distributions and tested on individual distributions.
>
>
> **Response to Individual Questions.**
>
>
>
>
>
> **(Q1)** *Your definition and narratives …  discuss more about your definition of ICL and how it relates to algorithm learning process.*
>
> Yes, indeed our definition of in-context learning is similar to those of [1, 2] which are also based on Garg et al [3]. There are no significant differences in our framework compared to [1] and our perspective of looking at the meta-learning-like setup as algorithmic learning is almost identical. Their [1] definition is more formal which is necessary for their theoretical results and not required to understand the main findings of our paper. Some of our experiments exploring the sample complexity of Transformers (Section D.3 and Figure 13) are also along the same lines as their [1] work.
>
> We are glad that you find the algorithmic learning perspective interesting. There are some theoretical insights about the representational capabilities of Transformers one can glean from such a perspective. Maybe you will find the following interesting: Recent works [5] have shown that hard-attention (hardmax instead of softmax) Transformers can represent functions in the class of AC0 circuits which are constant depth polynomial size Boolean circuits (with ‘and’ and ‘or’ gates). We found that we can represent the known PAC-learning algorithm for learning Conjunctions with a Boolean circuit in AC0 which implies that hard-attention Transformers are capable of representing the learning algorithm for Conjunctions (but does not imply that they can learn). If you find this interesting, we can discuss this in detail in our next version. Given space constraints we cannot include it in the main paper.

---

> ### Author Response · Authors · 2023-11-14
> **Response to Questions (Cont'd)**
>
> **(Q2)** *I am not sure how you sampled the random conjunction function for each task. … ?*
>
> The details of the distribution over functions are provided in Appendix C.2. Regarding your question about how Conjunctions are sampled, this is the description in the paper,
>
> > For tasks such as Conjunctions and Disjunctions which are either AND ($\wedge$) or OR ($\vee$) of some of the $2n$ literals, the function is sampled such that each literal $x_i$ or its complement $\bar{x}_i$ has a probability of $p$ in being in the sampled function. In other words, for each $i \in [n]$, the literal $x_i$ has $p/2$ probability of being sampled, the literal $\bar{x}_i$ has $p/2$ probability and with $1-p$ probability the literal or it's complement is not included. In our main experiments, we set $p=30\%$ and also investigate the robustness of the trained model when evaluated on other distributions (other values of $p$) in Appendix D.1.
>
> In other words, the number of variables is not fixed. The conjunction can also have negative literals as well. We understand that it could be difficult to find this description in the appendix and hence we will add an explicit label to this in the next version of the paper.
>
>
> **(Q3)** *Is it possible for you to show what algorithm the TFs actually implement on some simple tasks like conjunction or disjunction?  … These can be hard even an open problem, which is definitely not a requirement.*
>
>
> We spent some time working on it earlier: Inspection of attention weights did not lead to any useful insights. We attempted to adopt the Transformer circuit framework [4] to get a more interpretable model. That approach requires training a simple 1- or 2-layer attention-only network (without any MLPs, layer norm, etc.) and we found that such simple models were unable to reach very high accuracy (unlike the vanilla Transformer) and hence it did not seem useful to try and interpret them. Further, we also tried training hard-attention (with very high softmax temperature so that the weights are almost 0 or 1) Transformers so that the weights can be more interpretable but such models did not perform as well and hence were not useful to interpret. Unfortunately, interpreting the exact mechanism behind Transformers’ computation for these learning problems seems like a separate problem and solving it could help solve multiple other problems. However, we believe that even without the interpretability aspects the insights in our paper could be useful to the largely active community working on understanding in-context learning phenomenon.
>
>
>
> **(Q4)** *For some tasks in your paper, there should be existing algorithms to learn it, such as conjunction or disjunction. I am also wondering how does trained TFs compare to them? … Bayesian optimal estimators (or some strong, computationally efficient baseline on specific task, if any).*
>
> We conducted such experiments and the results are present in the paper.  Figure 9 in the paper compares the trained Transformers with known PAC-learning algorithm for tasks such as Conjunction and Disjunctions. There are a few things to note here: (a) These classical algorithms are near-optimal and are guaranteed to work for any distributions over the input. They are optimal in the sense that the lower bound on the sample complexity of learning Conjunctions/Disjunctions is $O(n)$ and these PAC-learn these classes with $O(n)$ samples. (b) Transformers can perform better on the first few examples because they have knowledge about the distributions of inputs and functions since they are exposed to it in the meta-learning stage. So in that sense, they are closer to the Bayes-optimal-estimator than the classical algo in their behaviour. However, unlike the provable PAC-learning algorithms, the algorithms learned by Transformers are not guaranteed to work for arbitrary distributions. (c)  Both Transformers and the classical algo reach (near) perfect accuracy after observing almost the same number of $O(n)$ examples.
>
>
> Regarding the second part of the questions, we think the idea is interesting but we are not sure how feasible it would be to implement and compare with the Bayes optimal estimator within the discussion period. We will look into it as soon as we are done with the experiments requested by other reviewers which are relatively direct in terms of implementation.
>
>
> [1] Transformers as Algorithms: Generalization and Stability in In-context Learning
> [2] Trained Transformers Learn Linear Models In-Context
> [3]  What can transformers learn in-context? a case study of simple function classes.
> [4] "A Mathematical Framework for Transformer Circuits", Transformer Circuits Thread, 2021.
> [5] Masked Hard-Attention Transformers and Boolean RASP Recognize Exactly the Star-Free Languages

---

> ### Author Response · Authors · 2023-11-21
> **Interpretability experiments and other additions**
>
> We have added new experiments exploring the interpretability aspect suggested in your review. In simplified terms, we find that for Conjunctions, Transformers implement an algorithm which is similar in principle to a known PAC-learning algorithm for learning Conjunctions but it does not follow it exactly.
>
> **Simplified Description**. The classical algorithm for learning Conjunctions starts with a hypothesis with all literals in it and it iteratively removes literals based on positively labelled examples. We apply a simple approach to approximate the Conjunction applied by the Transformer model to make its prediction while learning in-context and observe a similar behaviour where it always starts with all literals and progressively drops literals as it sees many positive examples finally converging to the correct one and achieving perfect accuracy. Please see Section E.2 for more details.
>
>
> We have made some major changes in our previous and current revisions based on your suggestions:
> -   *Interpretability.* Section E.2: Exploring how Transformers learn conjunctions in-context and finding that their behaviour is similar in principle to a known algorithm.
> -   *Expressivity.* Section E.1: We add a result showing the circuit complexity of the known algorithm for learning Conjunctions which also leads to the result that even hard-attention Transformers can represent that algorithm.
> -   *Suggested Discussions.* Section E: Discuss the perspective of viewing the setup as finding learning algorithms in greater detail and discuss Transformers' performance relative to the classical algorithm for learning Conjunctions.
>
> We have also made some other minor changes to improve clarity based on your comments. Please see the general response for more details.
>
> We hope this has helped address your concerns. In light of our response and the new experiments we have conducted, we would like to request you to kindly consider increasing the score.

---

> ### Author Response · Authors · 2023-11-22
>
> We wish to emphasize that the author-reviewer discussion period ends today.  We have made major revisions based on your comments and provided detailed responses to your questions. Hence, we kindly request you to please check the revisions and rebuttal and consider adjusting your score if your concerns are addressed.

---

### Official Review · Reviewer_tbmK · 2023-11-01

**Soundness:** 3 good
**Presentation:** 2 fair
**Contribution:** 2 fair
**Rating:** 6
**Confidence:** 4

**Summary:**

This paper studies Transformer's abilities to learn boolean functions. The main results are three-folds:

- **Transformer's performance** on the boolean functions can be grouped into 3 categories:
  - Perfect: this include conjunctions, (sparse) disjunctions, CNFs and DNFs, and nearest neighbors.
  - Above random but not perfect: this include majority, threshold, and integer halfspace.
  - Random-level: (sparse) parity. In particular, for Parity-(10,2), the fully connected network can learn to be perfect while Transformers are at chance-level.

  The paper also checks the performance of difference models, including LSTM, DSS (a state-space model), Hyena (a long convolutional model), RetNet (a hybrid model). These models are all worse than Transformers on the in-context learning of boolean functions, with Hyena being the closest.

- **Ability of learning from teaching sequences**: Transformers are able to leverage these teaching sequences well and learn the task with significantly fewer samples.
  -  Teaching sequences refer to sequences of samples that are sufficient to uniquely determine the function from a given function class.

- **Ability of pretrained models**: the paper studies two variants, 1) GPT2 with trained embedding and decoding layers (i.e. treating each $\{0,1\}^n$ as a sample to be embedded), and 2) direct evaluation of LLMs (i.e. treating $\{0,1\}^n$ as a sequence of 0,1 tokens). Transformers perform reasonably well in both setups. For the GPT2 experiments, the authors identified attention heads that closely implement nearest neighbors, similar to induction heads.

**Strengths:**

- The boolean evaluation setup is clean and controllable.
- The paper presents a large set of experiments.

**Weaknesses:**

- The takeaway messages are a bit unclear to me. [_Update_: The authors have clarified in the rebuttal. I hope the changes could be reflected in the revised paper.]
- Some results could be better presented and explained.

**Questions:**

- Parity-20: what if reporting the accuracy on every position, rather than the final position?
  - i.e. whether the full-sequence accuracy is too harsh, and some more continuous progress measure might be more informative.
- For OOD test results (Fig 10), how about the comparison to LSTM, especially under aggressive distribution shift where Transformers fail to preserve the in-distribution accuracy?

- About teaching sequences: [_Update_: the questions have been addressed during the rebuttal. I hope the updated paper could include the clarifications in the discussions; for example, which specific aspects of the empirical results suggest "two algorithms", and that it's important for the teaching sequences to be at the beginning of the list of examples, rather than mixed in as an arbitrary order. ]
    - Fig 2: what's the value of $k$? To better show the progression of performance, could you please start the x-axis from 1?
    - I'm curious about the robustness of the teaching sequence results. For example, for different values of $k$ (with dimension fixed to 20), does Transformer always able to
    - For the teaching sequence experiments, I wonder what would happen if during test time, we permute the $t$ teaching sequence samples, or permute all $m$ samples.
    - Appendix F: why would this not work as a teaching sequence for the entire set of parity?
        - Note that $k$ samples suffice only if the value of $k$ is known, and this would imply that the shortest teaching sequence for the full set of parity is of length 0.
    - Relatedly, the point that "FFN cannot learn from only the teaching sequences" seems to be naturally true: the number of samples required in the teaching sequence is defined _given that the function class is known_. For example for sparse parity, $k$ samples suffice only if we know that the task is $k$-sparse parity and we know the value of $k$. However, the function class is not specified to the FFN, hence it's expected that $k$ samples alone are insufficient to learn the task.
    - I'm not sure it's fair to say that Transformers learn "two distinct algorithms" for learning with and without teaching sequences, since this seems to me as simply a matter of distribution shift: if the Transformer has only seen sequences where the first $t$ samples are from a teaching sequence, then there is a drastic distribution shift in the test time when there's no such samples. What am I missing here?

Misc comments
- Section 5: the paper says real-value functions are not straightforward to evaluate "since LLMs receive and produce discrete values": I'm not sure how much this distinction is true or relevant, since LLMs do have embeddings which are vectors of real values.
- Table 1: for better readability, perhaps consider highlighting the numbers with colors (e.g. set the text background according to some colormap) would make it easier to read.

---

> ### Author Response · Authors · 2023-11-14
> **Response to Weakness**
>
> Thank you for your thoughtful comments and time.
>
> Response to Weakness: “Reviewer: *The takeaway messages are a bit unclear to me.*”
>
> List of takeaways.
>
> **(Takeaway 1)** Transformers (as well as other architectures) have concrete limitations in in-context learning certain classes such as Parities which are known to be learnable in polynomial time. They even fail when provided with a number of examples which is sufficient for a feedforward network trained with gradient descent.
>
> While there are various classes of functions for which Transformers have been shown to perform well, we are not aware of works which indicate a precise class of functions (such as Parities) which are known to be efficiently learnable and Transformers fail to learn such classes of functions in-context.
>
> **(Takeaway 2)** Attention-free architectures can perform in-context learning as well (in the sense of implementing learning algorithms). However, there still exists a gap between their performance and Transformers’ performance. We are unaware of previous works that report such a comparison on in-context learning with such a benchmark of learning problems. Reviewer bf3N found both takeaways 1 and 2 interesting and noted that they can be of interest to the community.
>
> **(Takeaway 3)** Transformers can leverage more informative examples such as teaching sequences to learn more sample-efficiently when such prompts are provided and can switch to the vanilla version when teaching sequences are not provided.
> Your specific questions about takeaway 3 are discussed separately.
>
> **(Takeaway 4)** Pretrained models such as GPT-2 which are trained on natural language data encode mechanisms to achieve non-trivial performance on learning problems such as Conjunctions and can implement the nearest neighbour algorithm.
>
> **(Takeaway 5)** LLMs primarily pretrained on text data can take a sequence of inputs and labels in a manner similar to the meta-learning-like setup and perform as well as certain baselines such as the nearest neighbour algorithm. Evidence for 4 and 5 indicate that LLMs can learn from in-context examples alone apart from just indexing to tasks already seen during training. In other words, they can act as learning algorithms to some degree in a manner similar to the way we test models trained from scratch in the meta-learning-like setup.
>
> To our knowledge, our experiments with LLMs (takeaways 4 and 5) are the first to explore the ability of LLMs to act as learning algorithms and solve learning problems.  Reviewer 9RLW found the experiments with LLMs and teaching sequences quite interesting.
>
> As mentioned in the introduction of our paper, our work focuses on the following research questions and the insights above are a step towards answering those.
>
> From Section 1:
> (Takeaway 1) -> (a) What are the limits of the in-context learning ability of Transformers?
> (Takeaway 2) -> (b) Is the attention mechanism essential for in-context learning?
> (Takeaway 3) -> (c) Can Transformers exploit high-quality and informative examples to learn more efficiently?
> (Takeaways 4 and 5) -> (d) To what extent do LLMs that are not specifically trained for these tasks carry the capacity to implement non-trivial learning algorithms on in-context examples?
>
>
> We hope this helps in clarifying the key takeaways related to the primary research questions explored in the paper. Let us know if anything regarding them is unclear.

---

> ### Author Response · Authors · 2023-11-14
> **Response to Questions**
>
> “**(Q1)** Reviewer: *Parity-20: what if reporting the accuracy on every position, rather than the final position?*”
>
> Figure 6 depicts the accuracy at every position and not just the final position for almost all of the tasks. For Parity-(10, 2), you can see that the accuracy at every position is near chance level. The performance on Parity-(20, 3) and Parity-20 is almost identical, i.e. near chance level at every position. It was omitted since the plots were almost identical to Parity-(10, 2).
>
> “**(Q2)** Reviewer: For OOD test results (Fig 10), how about the comparison to LSTM, especially under aggressive distribution shift where Transformers fail to preserve the in-distribution accuracy?”
>
> We will try to do experiments along these lines and report back in a few days.
>
>
> “**(Q3a)** Reviewer: *Fig 2: what's the value of k? To better show the progression of performance, could you please start the x-axis from 1?*”
>
> The value of k is not fixed. It depends on the number of literals in the target Conjunctions or DNFs. For Figure 2 Left, the expected length of the teaching sequence in the Conjunction is ~27 based on the distribution of the class of functions. For function classes such as Conjunctions or DNFs, most of the examples in the teaching sequence have negative labels and hence any kind of baseline even Null accuracy can achieve near-perfect accuracy on the inputs in the teaching sequence or equivalently on the first 30 examples on average. However, the performances of the baselines drop right after the teaching sequence whereas Transformers stay at perfect accuracy on the rest of the examples. Hence, the performances on the first ~27 examples primarily containing teaching sequences are not informative of the models’ true effectiveness and were omitted to avoid confusion. Based on your comment, we will include the plot where the x-axis begins with 1 in the appendix along with the explanation of their performance in the next couple of days.
>
> “**(Q3b)** Reviewer: I'm curious about the robustness of the teaching sequence results. For example, for different values of  (with dimension fixed to 20), does Transformer always able to”
>
> For classes such as Conjunctions and DNFs, the length of the teaching sequence varies so the performance of Transformers is not specific to teaching sequences of a particular length. For Parity-(20, 3), the length of the teaching sequence is fixed since the output always depends on a fixed number of variables in sparse Parities. If we change the length of the teaching sequence, then the class of functions will change as well and hence it becomes more of a problem about generalizing to unseen functions apart from generalizing to teaching sequences of different lengths.
>
> “**(Q3c)** Reviewer: *For the teaching sequence experiments, I wonder what would happen if during test time, we permute the t teaching sequence samples, or permute all m samples.*”
>
> We actually do permute the examples in the teaching sequences for the experiments with teaching sequences.
>
> We have not explored the setting where all the examples in the prompt are permuted since it is not directly related to the motivating research questions of our work. However, we are conducting such experiments and will report the results in the next few days.
>
> “**(Q3d)** Reviewer: *Appendix F: why would this not work as a teaching sequence for the entire set of parity?
> Note that k samples suffice only if the value of k is known, and this would imply that the shortest teaching sequence for the full set of parity is of length 0.*”
>
> That is not correct. We think you may have misinterpreted the definition of the full set of Parities. The statement you are referring to is the following,
>
> > Page 24: “This would not work as a teaching sequence for the entire set of Parity functions Parity-n.”
>
> For Parity-(n, k), the function value depends on exactly k bits and the full set of Parities for any n (Parity-n) contains all such functions (for all k). Quoting the description from the paper,
>
> > Page 4: “The class Parity-n contains all $2^n$ possible Parity functions over $\{0, 1\}^n$. We denote the class of sparse parities with Parity-(n, k) which contain functions with $k$ relevant variables defined over $\{0, 1\}^n$.”
>
> What you said would have been true if the full set of Parities would have had only one function which depends on all bits (Parity-(n, n)).

---

> ### Author Response · Authors · 2023-11-14
> **Response to Questions (Cont'd)**
>
> “**(Q3e)** Reviewer: *Relatedly, the point that "FFN cannot learn from only the teaching sequences" ….  insufficient to learn the task.*”
>
> That is right and since FFNs+SGD are general-purpose learning algorithms it is very difficult to integrate such problem-specific knowledge so that they perform more optimally for a specific problem.
>
> The claim here is not that Transformers are better general-purpose learning algorithms than FFNs+SGD. To us, it is interesting to observe that Transformers can learn near-optimal task-specific (vanilla vs teaching seqs) learning algorithms and depending on the input distribution during test (whether it contains a teaching seq or not), they can perform the algorithm that is more optimal for the particular input. So in some sense, where task-specific knowledge can be leveraged, they can perform more optimally than FFNs+SGD in some cases. This is true for other task-specific learning algorithms as well. For instance, Gaussian elimination is far more efficient for learning parities than with FFN+SGD but they are task-specific. While models such as Transformers can leverage problem-specific knowledge in some cases, it is clear that they fail on tasks such as learning sparse parities where FFNs+SGD can succeed in the learning problem.
>
> We will add a footnote to that statement clarifying that the claim is not that Transformers are better general-purpose learning algorithms than FFNs+SGD.
>
>
>
>
>
>
>
> “**(Q3f)** Reviewer: *I'm not sure it's fair to say that Transformers learn "two distinct algorithms" for learning with and without teaching sequences, since this seems to me as simply a matter of distribution shift: …*”
>
> When we test Transformers trained for vanilla Conjunctions on examples with teaching sequences, there is a distribution shift as well but they still perform well (almost in the same way as without the distribution shift). Ignoring Transformers for now, one can consider two types of algorithms for learning Conjunctions. Algo (a): a general purpose algo such as an FFN trained with gradient descent. Algo (b): The second algorithm is dependent on teaching sequences and uses the teaching sequence to identify the target conjunction.
>
> Note that the first algorithm can work with and without teaching sequences but it is natural to expect it to be suboptimal for inputs with teaching sequences. Similarly, the second algorithm will be optimal for inputs with a teaching sequence but will completely fail when the inputs do not have a teaching sequence. When we say that Transformers can learn two distinct algorithms, we mean it in this sense. From Figure 3 left and Table 1 (or Figure 6 top left), you can see that Transformers trained on vanilla Conjunctions behave like the first algorithm we described above. Similarly, from Figure 3 center-left and Figure 2 left, you can see that when trained with teaching sequences, they behave more like the second algorithm dependent on the teaching sequence. The emphasis here is on two different algorithms for the same class of functions (Conjunctions) where their effectiveness is dependent on the input distribution. If all aspects of the learning problem are identical then there is no reason to find two different algorithms.
>
>
>
>
>
>
> “**(Q4a)** Reviewer: *Section 5: the paper says real-value functions are not straightforward to evaluate "since LLMs receive and produce discrete values": I'm not sure how much this distinction is true or relevant, since LLMs do have embeddings which are vectors of real values.*”
>
> We do not agree and the distinction is quite true. While training a Transformer of width (or d_model) $d$ on real-valued regression problems with $n$ dimensional inputs to learn functions of the form $f:  \mathbb{R}^{n} \rightarrow  \mathbb{R}$,  there is a trainable linear map $W:  \mathbb{R}^{n} \rightarrow  \mathbb{R}^{d}$ that transforms a real-valued vector to the same space as the width of the model. Pretrained LLMs do not have this linear map and even if we can provide real-valued vectors of the same dimension as the width (or d_model) of the model, it does not seem reasonable to draw conclusions from its performance since it has never been trained on those vectors. While training Transformers on Boolean functions in the meta-learning-like setup there is a similar linear map as well  $W:  \{0, 1\}^{n} \rightarrow \mathbb{R}^{d}$, however since the inputs are discrete and there are separate embeddings for the tokens ‘0’ and ‘1’ in the LLM, we can test the models by providing an n-dimensional Boolean input with n different tokens. This is the setup with ‘Direct evaluation’ in Section 5.

---

> ### Comment · Reviewer_tbmK · 2023-11-18
> **Further clarifications**
>
> Thank you for the detailed responses and the clarifications! I have a few more questions which I hope to get the authors' clarifications on.
>
> **Further comments on the takeaways**
> - Takeaway 1: Transformers cannot learn parity efficiently: I think this has been known. Theoretically, parity is well-known to be statistically easy but computationally hard, so just stating the statistical complexity is not sufficient. Empirically, various prior results (see the two papers below) have shown that parity is hard for Transformers to learn even in standard training with SGD; given these results, I think it is expected that Transformer will also struggle to learn parity in-context.
>     - Bhattamishra et al. 20, On the Ability and Limitations of Transformers to Recognize Formal Languages.
>     - Liu et al. 23, Transformers Learn Shortcuts to Automata.
> - Takeaway 2: attention-free architecture can perform in-context learning: Xie et al. 21 already showed that LSTM has in-context learning abilities.
> - Takeaway 4 & 5 can be summarized as Transformers can meta-learn in-context; could you please elaborate how this contrasts with prior work such as Bai et al. 23?
>
> **Follow-up on the questions**
> - (Q3e) Regarding task-specific, there is a difference between "task-specific" and "algorithm-specific". I meant to say that conditioning on in-context examples provides some information of the task (not necessarily the algorithm), hence it's an unfair comparison if FFN hasn't been given the same type of conditioning (am I interpreting the FFN setting correctly here?).
> - (Q3f) I'd like to clarify which empirical evidence supports the claim of two different algorithms: are you referring to the qualitative difference that 1) training without teaching sequences has a gradual increase in performance as the number of examples increases, whereas 2) training with teaching sequences has a performance that does not vary much as the number of examples increases? I understand that the training distributions are different and that's why the learned models are different. What I'm trying to check is whether the learned models have sufficiently distinct behaviors to be considered as "two algorithms", as opposed to merely being a manifestation of OOD behaviors.

---

> > ### Author Response · Authors · 2023-11-19
> > **Response to comments on the takeaways**
> >
> > Thank you for your response.
> >
> > Takeaway 1: “*Transformers cannot … given these results, I think it is expected that Transformer will also struggle to learn parity in-context.*”
> >
> >
> > This comment is not correct. The word ‘Parity’ in the context of our paper and the works you cited [1, 2] have **different** meanings. In papers that you have cited ‘Parity’ refers to a single function (a two-state automaton) over variable length strings which outputs 1 if the number of 1s in all bits is odd. In the context of our work, Parity refers to a class of functions over fixed-length inputs where the output depends on a *subset* of bits. These are categorically different problems.
> >
> > It is untrue that “*various prior results [1, 2] have shown that parity is hard for Transformers to learn even in standard training*”. (a) For Parity tasks considered in our work (such as Parity-(10, 2), etc.), evidence from prior work [5] shows that they can learn that quite well and even better than LSTMs. Hence it is not obvious that they fail to learn them in in-context setting. (b) Additionally, for the Parity automaton problem, the results in [2] are in contradiction to your statement. They show that Transformers with reasonable depths (greater than 3 or 4) can generalize well in the in-distribution case (see Figure 3 a, b) in [2]). The limitations of Transformers on the Parity automaton problem are primarily concerned with generalizing to higher lengths which is unrelated to our problem.
> >
> >
> >
> > Takeaway 4 & 5 “*can be summarized as Transformers can meta-learn in-context; could you please elaborate how this contrasts with prior work such as Bai et al. 23?*”
> >
> > We would like to emphasize that Takeaways 4 and 5 have **no relation** to the results in Bai et al [4]. The results in Bai et al [4] are with Transformers trained from scratch in the meta-learning-like setup and not LLMs pretrained on natural language data such as GPT-2, LLaMA or GPT-4. Recall that Takeaways 4 and 5 (Section 5) are with actual pretrained models and not Transformers trained from scratch. For Takeaway 5, there is no training involved with the LLMs. For experiments in Takeaway 4, the parameters of the Transformer model in GPT-2 are frozen.
> >
> > As mentioned earlier, to our knowledge, our experiments with LLMs pretrained on natural language data among other things are the first to systematically explore the ability of LLMs to act as learning algorithms and solve learning problems in a manner similar to the artificial meta-learning-like setup.
> >
> >
> >
> > Takeaway 2: “*attention-free architecture can perform in-context learning: Xie et al. 21 already showed that LSTM has in-context learning abilities.*”
> >
> > We believe there is a significant difference between showing that LSTMs can perform a particular in-context learning task [3] and conducting a systematic study with multiple architectures on a test bed with a variety of tasks showing that (a) several attention-free variants (linear recurrent, long conv, etc) can perform in-context learning for multiple tasks, (b) each of these architectures has certain limitations in comparison with Transformers. We observe different strengths and limitations of each architecture.
> >
> > The takeaways from [3] and our work are different as well. Unlike the takeaway from [3], evidence from our work suggests that while attention may not be necessary and various attention-free models can solve multiple in-context learning tasks, they are not sufficient at the moment to match Transformers’ performance on all tasks considered in our work.
> >
> > One oversight on our part is that while we have cited Xie et al [3], we have not cited this aspect appropriately and we will rectify it in the next version.
> >
> > In our work, the experiments regarding takeaway 2 have a clear motivation: to systematically understand the performance of different types of attention-free models (state-space, long-convolution, recurrent) with respect to Transformers. Developing attention-free alternatives to Transformers is an active area of research at the moment. We argue that our results on a more systematic benchmark highlighting the limitations and effectiveness of different types of models could be informative to researchers working on developing more effective models.
> >
> >
> >
> > [1] On the Ability and Limitations of Transformers to Recognize Formal Languages.
> > [2] Transformers Learn Shortcuts to Automata.
> > [3] An Explanation of In-context Learning as Implicit Bayesian Inference.
> > [4] Transformers as Statisticians: Provable In-Context Learning with In-Context Algorithm Selection.
> > [5] Simplicity Bias in Transformers and their Ability to Learn Sparse Boolean Functions.

---

> > ### Author Response · Authors · 2023-11-19
> > **Response to follow-up questions**
> >
> > (Q3e) "*Regarding task-specific, there is a … of conditioning (am I interpreting the FFN setting correctly here?).*"
> >
> > We are not completely sure about your interpretation. The in-context examples can be viewed as a training set for Transformers (for in-context learning). Similarly, the same examples are provided as a training set to the FFN and optimized with gradient-based methods. In both cases, the goal of the examples is to provide information about the target labelling function. An algorithm dependent on the teaching sequence is task-specific in the sense that it requires the teaching sequence to be in the training set otherwise it is not guaranteed to work.
> >
> > The comparison with FFN is unfair for all the problems in Table 1 since the hypothesis space of FFNs is not restricted to the class of functions in the learning problem. In contrast, during the pretraining (or meta-learning) stage, Transformers (or other models) are provided with the observation that the inputs are always labelled by functions of a particular class (e.g. Threshold functions, Conjunctions, etc). Nonetheless, FFNs+GD still perform better on learning multiple tasks such as Threshold functions, Integer-halfspace and Sparse Parity. This indicates that the performance of Transformers (and other models) is suboptimal in those tasks. We do not disagree that comparing FFNs on training examples with teaching sequence is unfair as well but it just seems like a point that is worth clarifying and is hence presented as a note in the main paper and as a supporting result in the appendix.
> >
> >
> > (Q3f) *"I'd like to clarify which empirical evidence supports the … behaviors to be considered as "two algorithms", as opposed to merely being a manifestation of OOD behaviors."*
> >
> > We believe there is sufficient evidence to suggest that their behaviour is different. For Conjunctions, as mentioned earlier, you can explicitly construct two algorithms (a) one for general distribution over inputs or functions and (b) another which is dependent on teaching sequence. The qualitative difference is that for algorithm (a) (independent of teaching sequence), we are likely to observe a gradual increase in performance for arbitrary distributions over inputs (such as Figure 6 (top-left)) or with teaching sequences (Figure 3 left). In contrast, algorithm (b) can be guaranteed to achieve perfect accuracy right after observing the teaching sequence (Figure 2 left) but poor accuracy when such teaching examples are not provided (Figure 3 centre-left).
> >
> > Transformers trained on vanilla Conjunctions (which are analogous to case (a)) are quite robust to distribution shifts based on our experiments. They behave in the same way even if we change the distribution over functions or inputs significantly and even work the same way with teaching sequences (though not as optimal as (b) and need more examples apart from teaching sequences like FFN+GD). For Transformers trained with teaching sequence, their behaviour is consistent with the second type of algorithm (b). They do work on OOD data as long as the inputs have teaching sequences in the beginning. The teaching sequences can be longer or shorter than seen during training and similarly, the target Conjunctions can have more or less literals than seen during training but they behave like algorithm (b). Like algorithm (b) they fail to perform well when the examples do not begin with a teaching sequence.
> >
> >
> >
> > **Revisions.** Also, we conducted some experiments and added discussions around the performance of FFN+GD with teaching sequences based on your previous comments. You can refer to the general response for the list of changes in the revised version and we hope it helps address some of your concerns.

---

> ### Author Response · Authors · 2023-11-22
>
> We wish to emphasize that the author-reviewer discussion period ends today. We added multiple discussions and an experimental result in the paper based on your comments and provided detailed responses to your questions. Hence, we kindly request you to please check the revisions and responses and consider adjusting your score if your concerns are addressed.

---

> ### Comment · Reviewer_tbmK · 2023-11-22
>
> Thank you for the further discussions, and I apologize for my late reply!
>
> - **re learnability of parity**: what I meant is that it's been well known that there is a computational-statistical gap for parity and various prior work has shown empirical evidence on how Transformers struggle to optimize for parity, hence I don't agree that "We are not aware of any other precisely defined function classes where Transformers fail and which are known to be PAC-learnable in polynomial time." as stated in the reply to Reviewer fGea.
>     - Also as a minor clarification, re contradicting results of Liu et al. 23: I was referring to the optimization challenge pointed out in Liu et al. 23, e.g their Fig 3(b), and hence there is no contradiction.
> - **re LSTM/attention-free networks**: I agree with the authors that the results in their work are different from the empirical study you have in this work. What I meant is that it's inaccurate to claim "in-context learning hasn't been observed in non-Transformer models", and including the discussion you have in the response would be nice.
> - **re comparison between Bai et al.**, I understand the training from scratch vs pretraining distinction, but I think the more interesting distinction is that this work is trying to show the learning of different algorithm for the _same_ task, as you mentioned in your reply to Reviewer fGea which I missed previously -- apologies for the oversight on my end. The current comparison in the paper wasn't clear enough to me, and including these points would be helpful.
> - **re teaching sequences**: I'd like to get some further clarification please:
>     - The authors replied that "The teaching sequences can be longer or shorter than seen during training"; could you point me to the results in the paper please?
>     - I understand there are results on permuted examples (in Fig 20), which are however negative, i.e. Transformers fail to learn in this case. Is this evidence that TF is not successfully learning with teaching sequences?
>     - Also, have you tested on unseen teaching sequences? i.e. divide the set of teaching sequences into two disjoint sets, one used for training, and one used for testing; let's say each teaching sequence is at the beginning of the set of examples for both train and test.

---

> > ### Author Response · Authors · 2023-11-23
> >
> > “*re learnability of parity: what I meant is that it's been well known that parity is computationally hard to learn and various prior work has shown … d hence there is no contradiction*"
> >
> >
> > First, we would like to highlight that previous works [1, 2, 3] have shown that Transformers can learn functions of the form Parity-(n, k). Among the Boolean function classes we considered, Parity is the most difficult in some sense (as discussed in Section F) but it is still difficult to have preemptively predicted all architectures (not just Transformers) would fail in such a manner.
> >
> > The statement “it's been well known that parity is computationally hard to learn and various prior work has shown empirical evidence on how Transformers struggle to optimize for parity, hence I don't agree that” ignores a lot of aspects of the results presented in our work. To begin with, if they are computationally hard to learn then how have previous works [1, 2, 3] found that Transformers trained from scratch perform well on the task. There are differences in the complexity of learning Parity-n and Parity-(n, k). As discussed in Section F of the paper, the hardness of learning Parities is primarily associated with the results that learning Parity-n requires $2^{\Omega(n)}$ queries and learning Parity-(n, k) requires $n^{\Omega(k)}$ queries in some restricted models of learning. Note that if $k$ is very small then it is a computationally tractable problem and hence previous empirical results [1, 2, 3] are with Parity-(n, k) types of problems.
> >
> > In our experiments, you can note that the values of $n$ and $k$ are quite small (compared to those considered in earlier works) since the number of in-context examples is limited. Hence, FFNs+SGD succeed in solving the Parity-(10, 2) problem but no architecture is able to perform beyond chance-level accuracy in the in-context setting.
> >
> >
> >
> >
> > [1] Simplicity Bias in Transformers and their Ability to Learn Sparse Boolean Functions.
> > [2] Inductive Biases and Variable Creation in Self-Attention Mechanisms.
> > [3] Hidden Progress in Deep Learning: SGD Learns Parities Near the Computational Limit.
> >
> >
> > "*re LSTM/attention-free networks: I agree with the authors …in non-Transformer models", and including the discussion you have in the response would be nice.*"
> >
> > We agree and we will add the discussion in the draft in a couple of hours.
> >
> >
> > re comparison between Bai et al., I understand the training … The current comparison in the paper wasn't clear enough to me, and including these points would be helpful.
> >
> > Okay, we will add the discussion to the paper in a couple of hours.
> >
> >
> >
> > "*re teaching sequences: I'd like to get some further clarification please:
> > The authors replied ... in the paper please?*"
> >
> > The results are not present in the paper currently but we will include it in the final version. We checked for their OOD performance when we first observed that they could successfully learn from teaching sequences. We did not have any specific reason to include it in the paper earlier. While we have tried to include an extensive set of experiments in our paper, it is natural that some empirical results useful to answer specific questions are not preemptively added to the paper.
> >
> >
> > "*I understand there are results on permuted examples (in Fig 20), which ... this evidence that TF is not successfully learning with teaching sequences?*"
> >
> > No, the behaviour is quite expected as the setting becomes close to the standard one. The main point of the experiments with teaching sequences provided altogether in the beginning is to check whether the model can use the entire teaching sequence and predict with perfect accuracy for the following examples which are randomly sampled from some arbitrary distribution. When we permute all examples then there is no longer a teaching sequence in the sequence of in-context examples.
> >
> >
> > "*Also, have you tested on unseen teaching sequences? i.e. divide the set of teaching ... of the set of examples for both train and test.*"
> >
> > The teaching sequences in the training and test sets are almost always different. The teaching sequence is determined by the target function and even for say Conjunction, the probability of the same Conjunction appearing during both training and evaluation is exponentially low. For instance, for our experiments with teaching sequences on Conjunctions, the experiments involve input dimensions of 75 where there are $3^{75} > 10^{35}$ Conjunctions (and associated teaching sequences). Hence, even if the model observes a million Conjunctions during training, most of the target functions and teaching sequences during evaluation will be unseen.

---

> > > ### Comment · Reviewer_tbmK · 2023-11-23
> > >
> > > re learning parity: Thank you for the clarification! One thing I missed is that the number of samples used in the experiments is more than $n^k$, which is mentioned in Section F.
> > >
> > > Thank you also for the clarification on the teaching sequences. Most of my concerns have been addressed, hence I'm raising the score to recommend an acceptance.

---

> > ### Author Response · Authors · 2023-11-23
> >
> > We have updated the paper and added the discussions as per your suggestions. At the moment, the differences with the prior work are added to Section J since we will exceed the page limit if we directly add them to the main paper. If the paper is accepted, we will add a concise but clear description of those in the main paper as well.

---

### Official Review · Reviewer_fGea · 2023-11-02

**Soundness:** 3 good
**Presentation:** 3 good
**Contribution:** 3 good
**Rating:** 6
**Confidence:** 4

**Summary:**

This paper studies the ability of transformer models to perform various boolean functions. More specifically, the authors train from scratch transformers on Boolean functions and they observe that  the models can achieve very high accuracy for some of them, but for some other (parities) they correspond to a  random guess; then they add in the training process a prompt that uniquely identifies the function that the model is trying to learn and train in the same way. In this case the models are able to perform well given that the identifying sequence is given at test time in the first tokens of the prompt. Finally, they train in the same tasks with and without the identifier;  in this setting they observe that the transformer performs well when given the identifier and close to the random guess when is not. The paper also includes similar observations for pretrained transformer models. They either train the embedding layers of pretrained models like GPT2 and then prompt them, or they simply prompt GPT4.

**Strengths:**

The paper is in general well written and the authors have performed multiple experiments with different models and baselines. The setting examined has some benefits, as it is discrete and it can be transferred to prompting models by changing the embedding layers. The experiments are consistent with what other authors have observed.

**Weaknesses:**

I am not sure which is the message of this paper compared to previous work. It has already been observed that transformers can "choose" between algorithms in [1], while we already knew that there are some tasks that transformers perform well and some others that they don't.
I think that all the observations in this paper have been observed in different settings in other papers, thus I consider the contribution marginal.

[1]: Bai, Yu, et al. "Transformers as Statisticians: Provable In-Context Learning with In-Context Algorithm Selection." arXiv preprint arXiv:2306.04637 (2023).

**Questions:**

1. When the authors create the datasets, it seems like they always sample a function and $m$ Boolean inputs. At test time, is the prompt used also generated in the same way? Sampling one function an $m$ Boolean inputs?
My question is related to whether the authors observe that these models can ``generalize'' to the case that the sequence length is not the one that the models were trained on.

2. How exactly the authors prompted GPT4? Is it just a sequence of examples? It seems that in the supplemental material there is a folder prompts that contains some algorithms.

3. Could the authors specify, which are the insights that this paper provides compared to previous work?

---

> ### Author Response · Authors · 2023-11-14
> **Response to Weakness**
>
> Thank you for your thoughtful comments and time.
>
> Regarding weakness and Question 3.
>
> We regret that the set of new insights in the paper was not clear. We first list the set of new insights below and then discuss each insight in some detail. We believe the key insights have not been explored in past works. For certain insights (such as learning distinct algorithms for the same task) we discuss the differences to past work (such as [Bai et al 2023] that you highlighted). If you disagree with any of the claims made regarding novelty in the discussion below, then it would be helpful if you could cite the relevant papers.
>
> List of insights
>
> 1. Show concrete limitations of Transformers in in-context learning certain precise function classes which are known to be efficiently learnable.
>
> 2. Transformers can leverage more informative examples such as teaching sequences.
>
> 3. Attention-free architectures can perform such kind of in-context learning as well but they still lag behind on certain tasks.
>
> 4. LLMs primarily pretrained on text data such as LLaMA-2 and GPT-4 can learn new functions in-context akin to baselines such as the nearest neighbour algorithm.
>
> 5. There are mechanisms encoded in the weights of pre-trained models such as GPT-2 which enable it to perform as well as baselines on learning classes like Conjunctions and also implement the nearest neighbour algorithm in a setup which is very close to the meta-learning-like setup.
>
>
>
>
> **(Insight 1)** While prior works have demonstrated that Transformers can learn various classes of real-valued functions by implementing gradient descent or by acting as a Bayes optimal predictor, our finding provides evidence that there are classes of functions where they perform poorly and some classes such as Parities where they fail to perform beyond chance-level accuracy. This can be of interest to researchers in the community and it can be further explored by future work to understand why this happens (as noted by Reviewer bf3N). While it is natural to expect any model to have limitations, it is important to note that Parities are learnable in polynomial time with algorithms such as Gaussian elimination and for the problem of learning Parity-(10, 2) we provide the models with input points which are even sufficient for FFN+GD to solve the task. Hence, the problem itself is not unsolvable and it was to some degree surprising to us that Transformers as well as other models fail to solve this task.
>
> **Why new?** We are not aware of any other precisely defined function classes where Transformers fail and which are known to be PAC-learnable in polynomial time.
>
> **(Insight 2)** We show that various other architectures can match the performance of Transformers on various learning problems but there still exists a gap. Various architectures struggle on multiple tasks and Hyena is closest to Transformers in terms of performance falling behind only in the nearest neighbour problem. This could be of interest to the community (as noted by Reviewer bf3N) and could be useful for researchers working on developing new architectures that can match Transformers’ performance as LLMs.
>
> **Why new?** We are not aware of prior works that either compare the ability of recently proposed attention-free architectures to implement learning algorithms or works that even have a benchmark with a wide spectrum of performance for evaluating the ability of architectures to implement learning algorithms.

---

> ### Author Response · Authors · 2023-11-14
> **Response to Weakness (Cont'd)**
>
> **(Insight 3)** While all prior works have focused on sampling examples from uniform or normal distributions, our adoption of Boolean functions allows us to experiment with specialized sequences of input which are more informative. Our results indicate that models such as Transformers can achieve (almost) perfect accuracy on all examples after observing the teaching sequence and can learn two distinct algorithms to learn the same class of functions. Reviewer 9RLW found the experiments with teaching sequences particularly interesting.
>
>
> **Why new?** To our knowledge, prior works have not explored specialized input/training examples and have mostly focused on standard distributions.
>
> Regarding your comparison with Bai et al [2023]. There is a subtle difference but we agree there are similarities. In our related work, we have clearly stated, quote below,
>
> > “More recently, [Ahuja, et al 2023, Bai et al 2023]  explored (among other things) the efficacy of Transformers in learning mixtures of tasks in-context and showed that they could adaptively select appropriate algorithms for a given sequence of inputs and labels.”
>
> Prior works have shown that Transformers can choose different algorithms for different classes of functions (say linear + logistic regression) depending on the sequence of inputs and labels. In our case, the emphasis is on choosing between different algorithms for the ‘same class of functions’ (e.g. Conjunctions). Unlike previous works, the target labelling function is still a Conjunction and the vanilla algorithm leads to near-perfect accuracy as well but Transformers are still able to choose the more sample-efficient algorithm which leads to near-perfect accuracy with fewer input examples.
>
>
> **(Insight 4)** Our results in Section 5 show that LLMs can learn new functions to some degree. All prior works on the meta-learning-like setup focus on Transformers trained from scratch and we believe it is imperative to understand if actual text-pretrained LLMs can learn to predict from such sequences of labelled examples as well. If they cannot, then it is unclear if any of the prior studies on the meta-learning-like setup have any implications on ‘In-context learning’. Our results provide new insights which demonstrate that LLMs such as LLaMA and GPT-4 can learn to predict as accurately as nearest neighbour baselines in various cases.
>
> **Why new?** Our results indicate that LLMs’ in-context learning ability goes beyond simply indexing from the set of tasks seen during training as suggested by some previous works  [Min, et al. 2022]. Prior works have argued that LLMs do not learn novel functions and simply recognize the function already seen during training when in-context learning tasks such as sentiment classification. Other works [Wies, et al. 2023] have developed theoretical frameworks for understanding LLM’s in-context learning ability based on such assumptions. Our results provide evidence against this and show that they can learn to predict unseen functions from in-context examples in certain cases.
>
> **(Insight 5)** GPT-2 can perform as well as baselines on learning Conjunctions and can implement the nearest neighbour algorithm in a setup which is very close to the meta-learning-like setup. It is important to note that GPT-2 is primarily pretrained on natural language data and not pretrained on any task that is close to this meta-learning-like setup but at the same time can achieve non-trivial accuracy on these tasks. Reviewer 9RLW found the experiments with LLMs interesting as well.
>
> **Why new?** To our knowledge, our results (insights 4 and 5) are the first evidence to indicate that LLMs can approximately learn new functions in a manner similar to the meta-learning-like setup.
>
>
>
>
>
> [Bai, et al.] "Transformers as Statisticians: Provable In-Context Learning with In-Context Algorithm Selection." arXiv preprint arXiv:2306.04637 (2023).
>
> [Ahuja, et al] "In-Context Learning through the Bayesian Prism." arXiv preprint arXiv:2306.04891 (2023).
>
> [Min, et al.] "Rethinking the role of demonstrations: What makes in-context learning work?." arXiv preprint arXiv:2202.12837 (2022).
>
> [Wies, et al.] "The learnability of in-context learning." Neurips 2023.

---

> ### Author Response · Authors · 2023-11-14
> **Response to Questions**
>
> “**(Q1)** Reviewer: *When the authors create the datasets, it seems like they always sample a function and m Boolean inputs. At test time, is the prompt used also generated in the same way? Sampling one function an m Boolean inputs? My question is related to whether the authors observe that these models can ``generalize'' to the case that the sequence length is not the one that the models were trained on.*”
>
> Yes during both training and evaluation, each prompt is created by sampling a function and $m$ input points. In some scenarios, the distribution over functions or input could be different but we have not conducted experiments to check if models can generalize to higher lengths during evaluation. We think most prior works have focused on this since Pretrained LLMs typically have a maximum context length (such as 4k, 8k, etc) and in-context learning is typically done within that.
>
> Having said that, we are conducting some experiments based on your question to test the ability to generalize to higher lengths and will report the results within the next few days.
>
>
>
>
> “**(Q2)** Reviewer: *How exactly the authors prompted GPT4? Is it just a sequence of examples? It seems that in the supplemental material there is a folder prompts that contains some algorithms.*”
>
> The details of the prompt are explained in Section G.3 and an example prompt is provided in Figure 17. Quoting a part from Section G.3 in the paper
>
> > The main prompt (see Figure 17 for an example) comprises of an instruction in natural language followed by a series of $k$ example points ($x_i, y_i$), where $x_i \in \{0, 1\}^d$ is provided as a sequence of tokens $x_1, \ldots, x_d$, $x_j \in \{0, 1\}$ and $y_i \in \{0, 1\}$ is also provided as a single token. The model's goal is to predict the correct label $y_{k+1} \in \{0, 1\}$ for the query point $x_{k+1}$. For a particular function, we sample a sequence of $n$ points and prompt the model with $k < n$ points as in-context exemplars of the function. We call the model multiple times, increasing $k$ by a value of 5 every time.

---

> ### Author Response · Authors · 2023-11-19
>
> We have conducted the additional experiment on length generalization (please check the general response) as suggested in your review and responded to your comments in our rebuttal. Could you kindly check them and let us know if your concerns are now adequately addressed?

---

> ### Author Response · Authors · 2023-11-22
>
> We wish to emphasize that the author-reviewer discussion period ends today. We added the experiment suggested by you in the paper and provided detailed responses to your questions. Hence, we kindly request you to please check the revisions and rebuttal and consider adjusting your score if your concerns are addressed.

---

> > ### Comment · Reviewer_fGea · 2023-11-23
> > **Response to authors**
> >
> > I would like to thank the authors for their responses, I have read the rebuttal of the authors and their responses to the rest of the reviewers. I understand better now the contribution of the paper. Considering my initial comments, I am sorry if they were somehow abstract. I was referring to results like the ones pointed out by Reviewer tbmK and various other results that TFs can learn linear regression, ridge regression etc.
> >
> > After the rebuttal I have raised my score accordingly.

---

### Official Review · Reviewer_bf3N · 2023-11-08

**Soundness:** 4 excellent
**Presentation:** 4 excellent
**Contribution:** 4 excellent
**Rating:** 8
**Confidence:** 4

**Summary:**

The paper investigates the task of training models to learn various boolean function classes in-context. While the prior work in this theme has mainly focussed on the Transformer architectures and real-valued functions (e.g. linear functions), this paper studies the performance of various architectures (LSTM, Hyena, a state space model etc.) and mainly focuses on boolean functions. Some notable results include:

1) On all function classes where Transformers can learn in-context, other architectures considered also work. However, the performance of other architectures can be worse. For instance, for the nearest neighbor function class, Transformers achieve 100% accuracy whereas other architectures achieve an accuracy of around 90% or less. The Hyena architecture particularly stands out and matches the performance of Transformers for all function classes considered except the nearest neighbors function class.

2) No architecture is able to perform better than chance on the task of in-context learning parities and sparse parities. This happens even in the setting where a feedforward neural network (trained on the in-context examples using gradient descent) is able to learn perfectly.

3) The paper also tests existing LLMs like Llama and GPT-4 (that are not trained explicitly for in-context learning) on in-context learning tasks in small dimensions which are guaranteed to not be in the training set, and shows that these models perform at par with a nearest-neighbors baseline.

**Strengths:**

I enjoyed reading this paper. There are two results that I mentioned in the summary that stand out for me:

1) No architecture considered is able to in-context learn parity. The problem instances considered can be solved perfectly with Gaussian elimination. This suggests that these models struggle to learn Gaussian elimination. This is the first interesting negative result that I know of in the in-context learning ability of these models (when explicitly trained for in-context learning), and deserves more investigation.

2) The comparison between different architectures and their in-context learning performance on various function classes also seems interesting. Why do all architectures except Transformers struggle to some extent on Nearest Neighbors? Why do RetNets and state space models perform worse on 0-1 Threshold functions? Trying to find answers to such questions can shed light on the role of different architecture components.

In general, the paper is full of nicely executed experiments and many of them would be interesting to the community.

**Weaknesses:**

I don't see any major weakness.

**Questions:**

Some questions/suggestions:

1) It seems all the architectures considered (except LSTMs) have the number of hidden layers and latent dimensions in the same range. It would be good to include details of the number of parameters or compute used for each architecture so as to ensure that there is some normalizing factor among the architectures considered. This is to make sure that architectures like LSTMs and RetNets are not performing worse due to having a substantially smaller parameter count.

2) It would be good to clarify in the Results section that the negative results only hold for the particular hyperparameter choices. In particular, these experiments don't rule out the possibility of substantially bigger models or models trained for much longer being able to perform better in certain cases.

3) In the abstract, it is mentioned that Transformers can learn gradient-based learning algorithms. The prior works that I know of either claim that Transformers can represent such algorithms or that their performance matches gradient-based algorithms - none of these imply that the trained Transformers actually encode gradient-based learning algorithms. This line should be rephrased to avoid confusion.

4) Here is a blog post that might be worth mentioning in the context of in-context learning abilities of real LLMs: https://www.alignmentforum.org/posts/c2RzFadrxkzyRAFXa/who-models-the-models-that-model-models-an-exploration-of

5) Did you explore using a curriculum for training model to in-context learn parities (similar to Garg et. al.)?

---

> ### Author Response · Authors · 2023-11-14
> **Responses to Questions**
>
> Thank you for your thoughtful comments and time.
>
> Responses to the individual questions below.
>
>
> “**(Q1)** Reviewer: *It seems all the architectures considered (except LSTMs) have the number of hidden layers and latent dimensions in the same range. It would be good to include details of the number of parameters or compute used for each architecture so as to ensure that there is some normalizing factor among the architectures considered. This is to make sure that architectures like LSTMs and RetNets are not performing worse due to having a substantially smaller parameter count.*”
>
> For all architectures apart from LSTMs, the parameter counts are almost the same since in architectures such as RetNet, Hyena, and DSS, only the attention layer is replaced with a different mechanism. The feedforward block has the same number of parameters which dominates the parameter count and the replacements for attention with Retention/LongConv also have projection matrices. LSTMs typically have a smaller parameter count because deeper LSTMs are more difficult to train and do not perform well. For instance in Figure 14 (right), you can see the generalization error of LSTMs get poorer for a larger number of layers.
>
> In the next version (in a few days), we will specify the largest model (in terms of parameters) we trained for each architecture. We hope that suffices.
>
>
>
> “**(Q2)** Reviewer: *It would be good to clarify in the Results section that the negative results only hold for the particular hyperparameter choices. In particular, these experiments don't rule out the possibility of substantially bigger models or models trained for much longer being able to perform better in certain cases.*”
>
> We agree and this holds for negative results in any paper which is supported by empirical evidence. As authors, our goal has been to be as extensive as possible in terms of experiments and hyperparameter tuning. Nonetheless, the scenario you mentioned can always occur for such studies.  We will add a note in the paper mentioning this.
>
>
>
> “**(Q3)** Reviewer: *In the abstract, it is mentioned that Transformers can learn gradient-based learning algorithms. The prior works that I know of either claim that Transformers can represent such algorithms or that their performance matches gradient-based algorithms - none of these imply that the trained Transformers actually encode gradient-based learning algorithms. This line should be rephrased to avoid confusion.*”
>
> We think this is not entirely correct. While [Garg, et al.] showed that Transformers can match the performance of gradient-based algorithms and [ Akyürek, et al.] showed that Transformers can represent such algorithms, later [Von Oswald, et al.] showed quite convincingly that Transformers learn to encode gradient-based algorithms. In particular, they provided a much simpler construction to show how Transformers can implement a gradient-based algorithm for linear regression and then they inspected the weight/parameters of a trained Transformer model and found that they encode the same mechanism as their construction (apart from some scaling corrections) (See for instance Figure 9 in their paper). In other words, their results seem quite convincing to support their main claim that Transformers can learn gradient-based learning algorithms.
>
>
>
> “**(Q4)** Reviewer: *Here is a blog post that might be worth mentioning in the context of in-context learning abilities of real LLMs:* https://www.alignmentforum.org/posts/c2RzFadrxkzyRAFXa/who-models-the-models-that-model-models-an-exploration-of”
>
> Thank you for the suggestion. We were not aware of this and will cite this in our next version.
>
> “**(Q5)** Reviewer: *Did you explore using a curriculum for training model to in-context learn parities (similar to Garg et. al.)?*”
>
> We ran some preliminary experiments earlier and found that it did not help improve the performance. We will conduct them again and include the results in the next version of the paper.
>
>
> [Garg, et al.] "What can transformers learn in-context? a case study of simple function classes."  Neurips 2022.
>
> [ Akyürek, et al.]  "What learning algorithm is in-context learning? investigations with linear models." ICLR 2023.
>
> [Von Oswald, et al.] "Transformers learn in-context by gradient descent." ICML 2023.

---

> > ### Comment · Reviewer_bf3N · 2023-11-21
> >
> > I thank the authors for the detailed reply and for running the curriculum learning experiment.
> >
> > Regarding Transformers learning gradient based algorithms,  Oswald et. al. show this in the case with linear attention layers without any MLPs. In fact, a recent work (https://arxiv.org/abs/2310.17086) argues that the performance of Transformers is closer to higher order methods than it is to gradient descent. In this sense, I don't think it is clear that Transformers (with MLPs) actually encode gradient descent. I think it would be good to rephrase that sentence in the abstract to avoid confusion. Anyway it is not related to the central claim of the paper.

---

> > > ### Author Response · Authors · 2023-11-22
> > >
> > > Thank you for your response. Based on your suggestion, we will amend the line in the abstract to avoid confusion.

---

### Author Response · Authors · 2023-11-14
**General Response**

We thank all the reviewers for their thoughtful feedback and their time. We are encouraged to see that they found the insights in our paper interesting  (Rev. bf3N, 9RLW), and found the experiments to be well executed (Rev. bf3N, 9RLW),  sufficient (Rev. 9RLW) and to be of interest to the community (Rev. bf3N). In this work, we demonstrated limitations of Transformers in in-context learning certain precise classes of functions which are known to be efficiently learnable. Further, we compared various attention-free architectures and found that they match Transformers’ performance on various tasks but a gap still exists. Reviewer bf3N found these insights particularly interesting. We also explored the ability of Transformers to learn from more informative examples such as teaching sequences. Lastly, in two different setups close to the meta-learning-like setup, we showed that LLMs can perform as well as baselines such as the nearest neighbour algorithm. Reviewer 9RLW found these experiments particularly interesting.

Reviewer fGea mentioned that the insights in the paper are not novel as the primary weakness of the work. We respectfully disagree and without citations, it is hard to respond to that. They provide one citation which is already appropriately cited in our paper and it is related to a result which is not a part of the central research questions of our paper. In individual responses, we have discussed the key insights of our paper in detail and we do not believe those insights are present in prior works.

Reviewer tbmK mentioned the takeaways from the paper being unclear as the main weakness and had a set of questions. We have discussed the weaknesses and specific questions by every reviewer in the individual responses.

Based on the reviewers’ comments, we are running further experiments and in the next few days, we will report them and make certain changes to the draft to incorporate the feedback from the reviewers.

---

> ### Author Response · Authors · 2023-11-17
> **Experiments and changes based on reviewers' comments**
>
> We have updated the draft to include new experiments, discussion and some other minor changes based on the reviewers' comments. Here is the summary of the changes.
>
> **Primary Changes:** Experiments and Results
> -   *Length Generalization experiment* **(R-fGea**): We explored the performance of models on prompts with more examples (length) than seen during training. TLDR: We find Transformers without positional encodings and just causal masking generalize well whereas models with absolute positional encodings struggle. Described in Section D.1 and Figure 12.
> -   *Permuted Example Prompts with Teaching Seq* (**R-tbmK**): We evaluate the performance of models on prompts containing a permutation of random examples and teaching seq examples. The results and setup are described in Section G.1 and Figure 19.
> -   *Transformers as Learning Algorithms* (**R-9RLW**): We discuss the perspective of viewing this setup as finding learning algorithms in greater detail. More importantly, we add a result showing the circuit complexity of a known algorithm for learning Conjunctions and discuss its implications on Transformers' expressivity. Described in Section E.
> -  *Curriculum learning for Parities* (**R-bf3N**): We discuss the experiments with curriculum learning for Sparse Parities in Section F.
>
> **Secondary Changes:** Minor/Writing
> -   Added additional plots related to Teaching Sequence experiments (Section G.1) (R-tbmK)
> -   Discussion on the performance of FFNs on teaching seq in Sec G.1 and clarification note in the main paper. (R-tbmK)
> -   Discussion on the performance comparison between Transformers and known PAC-learning algorithm for Conjunctions (R-9RLW).
> -   Minor clarifications based on comments from R-bf3N and R-9RLW.
>
>
> We hope that the new results and draft changes have helped address the reviewers' concerns.
>
> Let us know if you need clarifications regarding the changes or our response to the individual reviews.

---

### Meta-Review · Area_Chair_4BMj · 2023-12-05

**Metareview:**

This paper provides a thorough empirical study on the in-context learning (ICL) ability of transformers for learning boolean functions, adding to the line of recent works that study ICL with transformers on certain simple function classes.

Apart from the setting being different from existing works (learning boolean functions) and demonstrating the amazing ICL capabilities of transformers, this work provides several additional interesting results on (1) comparison between transformers and other sequence architectures and baselines; (2) a finding about the limitations of transformers (on learning parity functions); (3) a study of the in-context sample efficiency and algorithm selection phenomenon through the "teaching sequence" settings; (4) A study of ICL in LLMs used in practice (such as GPT-4 and LLaMa-2), both with trainable embedding functions and directly through prompting. I find these contributions highly valuable to the line of works on ICL, addressing some of the most common questions people have with this line of results, and having the potentials to motivate many future studies. I congratulate the authors for the nice work.

The reviewers initially had some concerns; however the majority of them were addressed by the rebuttal and discussions.

**Justification For Why Not Higher Score:**

N/A

**Justification For Why Not Lower Score:**

This paper could be of wide interest to the community on understanding in-context learning and transformers in general.

---

### Decision · Program_Chairs · 2024-01-16

Accept (oral)